# Information Estimation with Discrete Diffusion

**Alberto Foresti, Giulio Franzese & Pietro Michiardi**
Department of Data Science
EURECOM
450 Route des Chappes, 06410 Biot, France
{alberto.foresti,giulio.franzese,pietro.michiardi}@eurecom.fr

## Abstract

Information-theoretic measures, such as Mutual Information (MI), play a crucial role in understanding non-linear relationships between random variables and are widely used across scientific disciplines. Yet, their use on real-world discrete data remains challenging. Existing methods typically rely on embedding discrete data into a continuous space and apply neural estimators originally designed for continuous distributions. This process requires careful engineering for both the embedding model and estimator architecture, but suffers from issues related to high data dimensionality. In this work, we introduce INFO-SEDD, a discrete diffusion–based approach that bridges information-theoretic estimation and generative modeling such that they can be used to compute Kullback-Leibler divergences. Backed by Continuous Time Markov Chains theory principles, the design of INFO-SEDD is lightweight and scalable and allows seamless integration with pretrained models. We showcase the versatility of our approach through applications on motif discovery in genetic promoter data, semantic-aware model selection in text summarization, and entropy estimation in Ising models. Finally, we construct consistency tests on real-world textual and genomics data. Our experiments demonstrate that INFO-SEDD outperforms alternatives that rely on the "embedding trick". Our results position INFO-SEDD as a robust and scalable tool for information-theoretic analysis of discrete data. We provide the open source code at https://github.com/AlbertoForesti/mutinfo-diffusion.

## 1 Introduction

Information theoretic measures represent a powerful tool to understand non-linear relationships between random variables (Shannon, 1948; MacKay, 2003) and find a wide range of applications in scientific fields (Karbowski, 2024; Eckford et al., 2016). Mutual information (MI), in particular, has become an established metric in machine learning (Bell and Sejnowski, 1995; Stratos, 2019; Belghazi et al., 2018; Oord et al., 2018; Hjelm et al., 2019), both for training models (Alemi et al., 2016; Chen et al., 2016; Zhao et al., 2018) and at inference time (Alemi and Fischer, 2018; Huang et al., 2020).

Estimating information theoretic quantities remains an open problem, and different paradigms for their estimation have emerged. Classical parametric and non-parametric methods (Pizer et al., 1987; Moon et al., 1995; Kraskov et al., 2004; Gao et al., 2015) have been recently superseded by variational approaches (Barber and Agakov, 2004; Nguyen et al., 2007; Nowozin et al., 2016; Poole et al., 2019; Wunder et al., 2021; Letizia et al., 2024; Federici et al., 2023) and neural estimators. Notably, several approaches rely on optimizing a lower bound (Papamakarios et al., 2017; Belghazi et al., 2018; Oord et al., 2018; Song and Ermon, 2019b; Rhodes et al., 2020; Letizia et al., 2025; Brekelmans et al., 2022). However, McAllester and Stratos (2020); Song and Ermon (2019a) challenge the applicability of this kind of estimator, especially in high MI scenarios, as they require an amount of samples that grows exponentially with the ground truth MI. To overcome this limitation, a new class of estimators has emerged (Franzese et al., 2024; Butakov et al., 2024; Kholkin et al., 2026). Unfortunately, these methods focus on continuous data and, despite their practical importance, few estimators for high-dimensional **discrete** distributions have been pro-

posed in the literature. While classical estimators for discrete random variables exist (Pinchas et al., 2024), their accuracy rapidly decreases with increasing data dimensionality. Applications that would benefit from scalable estimators of MI include, among others, DNA or peptide sequencing (Newcomb and Sayood, 2021; Xia et al.), text summarization (Darrin et al., 2024) and neuroscience (Chai et al., 2009), to name a few examples. Consequently, the development of new estimation techniques is of paramount importance for the broader scientific community.

A common workaround to deal with high-dimensional, discrete data is to embed it in a continuous space and use neural estimators conceived for continuous distributions. One recent example is (Lee and Rhee, 2024), where it is shown that the embeddings of pretrained language models can provide meaningful representations to estimate information theoretic quantities in unstructured data. However, such a process may not fully capture the discrete nature of the underlying data and might suffer from several limitations, such as the necessity to consider application-specific embeddings. For example, we show that MINDE (Franzese et al., 2023), which is a strong MI estimator in continuous domains, like image data, struggles with discrete data.

In this work, we build on the growing literature of diffusion-based estimators (Kong et al., 2022; Franzese et al., 2024), and present INFO-SEDD, a novel method for estimating information theoretic quantities of discrete data using Continuous Time Markov Chains (CTMCs) (Lou et al., 2024). These stochastic processes have recently seen a surge in popularity for applications such as generative language modeling (Lou et al., 2024; Sahoo et al., 2024; Nie et al., 2026). Their fundamental working principle is the reversal of a perturbation process which starts with clean data from a given distribution and that converges to uninformative noise. The workhorse of these approaches is the *score function*, which contains information about the probability distributions associated with the CTMCs at different time instants. Our proposed method, INFO-SEDD, builds upon such mathematical framework, extending it via Dynkin's lemma (Hanson, 2007), and leverages score functions to compute key information-theoretic metrics, such as MI between two random variables, and the entropy of a given distribution. By carefully selecting perturbation processes, our approach requires training only a single parametric model to compute MI across arbitrary subsets of variables. Furthermore, INFO-SEDD seamlessly integrates with pretrained models, without requiring ad-hoc procedures to work with discrete data.

To rigorously evaluate our method, we perform a large number of experiments both on synthetic and real data, and compare INFO-SEDD to state-of-the-art methods. We design a synthetic benchmark with ground truth MI that presents challenges such as high data dimensionality and high MI scenarios. Our results demonstrate that INFO-SEDD is both robust and consistently outperforms existing estimation methods. We then focus on two application domains in which we compare INFO-SEDD to competitors, which all require the "embedding trick" discussed above. First, we tackle the text summarization domain, evaluate the consistency of MI estimators, and study if MI represents a meaningful signal to perform model selection, as INFO-SEDD estimates are well aligned with human metrics for evaluating text summarization. We also present results in the domain of genomics, whereby we evaluate the consistency of MI estimators, and show that INFO-SEDD can be used to detect motif location in DNA sequencing data. Overall, our results indicate that INFO-SEDD outperforms alternatives for MI estimation, and pave the way to exploit information measures as meaningful signals for a variety of downstream tasks involving high-dimensional discrete data.

## 2 METHODOLOGY

We begin by defining the relation between CTMCs over discrete state spaces (Anderson, 2012) and the computation of Kullback-Leibler (KL) divergences. First, in Section 2.1, we provide a brief introduction to the fundamentals of CTMCs, emphasizing their time-reversal properties and parametric approximations (Lou et al., 2024). Then, in Section 2.2, we demonstrate how these processes can be adapted for divergence estimation, specifically by analyzing two processes that share the same generator but differ in their initial conditions.

### 2.1 PRELIMINARIES

Consider a CTMC $\overrightarrow{X}_t, t \in [0, T]$, defined over a finite state space $\chi = \{1, \ldots, N\}$ and specified by the infinitesimal generators $\overrightarrow{Q}_t : [0, T] \to \mathbb{R}^{N \times N}$, where the diagonal entries satisfy

$\overrightarrow{Q}_t(a,a) = -\sum_{a \neq b} \overrightarrow{Q}_t(a,b)$, with $\overrightarrow{Q}_t(a,b) \geq 0, a \neq b$. As established by Anderson (2012), the time evolution of the probability distribution $Pr(X_t = x) \stackrel{\text{def}}{=} \overrightarrow{p}_t(x)$ satisfies the following Ordinary Differential Equation (ODE): $\overrightarrow{p}_t = \overrightarrow{p}_0 + \int_0^t \overrightarrow{Q}_s \overrightarrow{p}_s ds$, where the initial conditions of the process $\overrightarrow{p}_0$ determine the distribution $\overrightarrow{p}_t$ at any time $t$. A key property of CTMCs is that their time-reversed counterpart, recently used for generative modeling purposes (Lou et al., 2024), also follows a CTMC, but with a different set of transition matrices. More precisely, defining the time-reversed process as $\overleftarrow{p}_t \stackrel{\text{def}}{=} \overrightarrow{p}_{T-t}$, the reverse process evolves according to the following ODE (Lou et al., 2024; Sun et al., 2023): $\overleftarrow{p}_t = \overrightarrow{p}_T + \int_0^t \overleftarrow{Q}_s \overleftarrow{p}_s ds$, where the reverse-time transition matrices $\overleftarrow{Q}_t$ relate to the forward transition matrices as follows:

$$\overleftarrow{Q}_t(b,a) = \left( \frac{\overrightarrow{p}_{T-t}(b)}{\overrightarrow{p}_{T-t}(a)} \overrightarrow{Q}_{T-t}(a,b) \right) (1 - \delta(a,b)) + \left( -\sum_{b \neq a} \overleftarrow{Q}_t(b,a) \right) \delta(a,a). \tag{1}$$

Under appropriate technical conditions on $\overrightarrow{Q}_t$ (Lou et al., 2024), the terminal distribution $\overrightarrow{p}_T$ converges to a known reference distribution $\pi$, which is independent of the initial distribution $\overrightarrow{p}_0$. This property enables sampling from $\overrightarrow{p}_0$ by simulating a CTMC with appropriately chosen generators (Sun et al., 2023; Kelly, 1981). However, other than simple and uninteresting scenarios, exact knowledge of the quantities $\frac{\overrightarrow{p}_{T-t}(b)}{\overrightarrow{p}_{T-t}(a)}$ is out of reach. A practical solution is to substitute in this numerical integration a parametric function $s_\theta^p(a,t)_b$, whose parameters are optimized according to the Diffusion Weighted Denoising Score Entropy (DWDSE) loss function (Equation 10, Section 3.2 of Lou et al. (2024)). Whenever the context is clear, we simplify the notation for the parametric score in the remainder of the paper and denote it with $s_\theta^p(\overrightarrow{X}_t)_x$ instead of $s_\theta^p(\overrightarrow{X}_t, t)_x$.

## 2.2 KL DIVERGENCES VIA CTMCS

Next, we show how to extend the CTMC framework to compute the KL divergence between two probability distributions $\overrightarrow{p}_0$ and $\overrightarrow{q}_0$ defined over the same support $\chi$, expressed as $\text{KL}\left[ \overrightarrow{p}_0 \parallel \overrightarrow{q}_0 \right]$. To achieve this, we construct two Markov chains that differ only in their initial conditions: one initialized from $\overrightarrow{p}_0$, and the other initialized from $\overrightarrow{q}_0$. The KL divergence can be expressed as:

$$\text{KL}\left[ \overrightarrow{p}_0 \parallel \overrightarrow{q}_0 \right] = \mathbb{E}\left[ \log \frac{\overrightarrow{p}_0}{\overrightarrow{q}_0}(\overleftarrow{X}_T) \right] = \mathbb{E}\left[ \log \frac{\overleftarrow{p}_T}{\overleftarrow{q}_T}(\overleftarrow{X}_T) \right] = \mathbb{E}\left[ \mathbb{E}\left[ \log \frac{\overleftarrow{p}_T}{\overleftarrow{q}_T}(\overleftarrow{X}_T) \Big| \overleftarrow{X}_0 \right] \right]. \tag{2}$$

The last term in Equation (2) can be rewritten using Dynkin's formula (Hanson, 2007), which states that for a generic function $f : \chi \times [0, T] \rightarrow \mathbb{R}$, we have:

$$\mathbb{E}\left[ f(\overleftarrow{X}_T, T) \Big| \overleftarrow{X}_0 \right] - f(\overleftarrow{X}_0, 0) = \mathbb{E}\left[ \int_0^T \frac{\partial f}{\partial t}(\overleftarrow{X}_t, t) + \mathcal{B}[f](\overleftarrow{X}_t, t) dt \Big| \overleftarrow{X}_0 \right], \tag{3}$$

where $\mathcal{B}$ is the *backward operator*, which we define as $\mathcal{B}[f](a,t) = \sum_{b \neq a} \overleftarrow{Q}_t(b,a)(f(b) - f(a))$.

By combining the result from Equation (3) with Equation (2), the KL divergence between discrete distributions can be conveniently approximated as:

$$\text{KL}\left[ \overrightarrow{p}_0 \parallel \overrightarrow{q}_0 \right] \approx \mathbb{E}\left[ \int_0^T \sum_{x \neq \overrightarrow{X}_t} \overrightarrow{Q}_t(\overrightarrow{X}_t, x) \left( K\left( \frac{\overrightarrow{p}_t(x)}{\overrightarrow{p}_t(\overrightarrow{X}_t)} \right) + \frac{\overrightarrow{q}_t(x)}{\overrightarrow{q}_t(\overrightarrow{X}_t)} - \frac{\overrightarrow{p}_t(x)}{\overrightarrow{p}_t(\overrightarrow{X}_t)} \log \frac{\overrightarrow{q}_t(x)}{\overrightarrow{q}_t(\overrightarrow{X}_t)} \right) dt \right], \tag{4}$$

where $K(a) = a(\log(a) - 1)$. We omit the term $\mathbb{E}\left[ \log \frac{\overleftarrow{p}_0}{\overleftarrow{q}_0}(\overleftarrow{X}_0) \right]$, as both $\overleftarrow{p}_0$ and $\overleftarrow{q}_0$ converge to $\pi$ (Lou et al., 2024).

The key quantities $\frac{\overrightarrow{p}_t(b)}{\overrightarrow{p}_t(a)}$ and $\frac{\overrightarrow{q}_t(b)}{\overrightarrow{q}_t(a)}$ in the expression above are not directly accessible. To address this, we substitute these ratios with parametric approximations optimized via DWDSE loss, leading to the following KL estimator:

$$\text{KL}\left[\overrightarrow{p}_0 \parallel \overrightarrow{q}_0\right] \approx \mathbb{E}\left[\int_0^T \sum_{x \neq \overrightarrow{X}_t} \overrightarrow{Q}_t(\overrightarrow{X}_t, x)\left(K\left(s_\theta^p(\overrightarrow{X}_t)_x\right) + s_\phi^q(\overrightarrow{X}_t)_x - s_\theta^p(\overrightarrow{X}_t)_x \log s_\phi^q(\overrightarrow{X}_t)_x\right) dt\right]. \tag{5}$$

Estimating Equation (5) using Monte Carlo techniques is straightforward: we sample time instants $t$ uniformly in $[0, T]$, simulate the forward process $\overrightarrow{X}_t$, and compute the required quantities using the parametric scores. Equation (5) enables estimating information measures, such as MI and Entropy. However, the KL estimator in its general form is not scalable for practical purposes: our practical method, described next, addresses this limitation.

## 3 OUR APPROACH: INFO-SEDD

We are now ready to introduce our MI estimator, INFO-SEDD, which is based on Equation (5). Given two random variables, $X$ and $Y$, we propose two approaches:

1. **Joint Method** [INFO-SEDD-J]: MI can be expressed in terms of the KL-divergence between the joint distribution $p_{XY}$ and the product of the marginals $m_{XY} = p_X \otimes p_Y$, $I(X, Y) = \text{KL}\left[p_{XY} \parallel m_{XY}\right]$, where $\otimes$ denotes the Kronecker product.

2. **Conditional Method** [INFO-SEDD-C]: alternatively, MI can be expressed as the KL divergence between the conditional distribution and the marginal distribution, $I(X, Y) = \mathbb{E}[\text{KL}\left[p_{Y|X} \parallel p_Y\right]]$.

In high dimensional applications, a naive implementation of Equation (5) quickly becomes unfeasible. Indeed, the size of the number of entries of the matrix $\overrightarrow{Q}_t$ scales with $|\chi|^2$, which is intractable. However, in many cases of interest, the random variables can be naturally decomposed into a structured sequence of $D$ subcomponents, each taking values from a discrete set of size $|\chi|$ (Lou et al., 2024; Austin et al., 2021; Campbell et al., 2022), i.e. $X = [X_1, \ldots, X_D]$, leading to a total state space of size $|\chi|^D$. This structured decomposition enables the use of sparse rate matrices, which constrain the CTMC to modify only one subcomponent at a time, significantly reducing computational complexity. This allows us to consider sequences which only differ in one component when computing the entries of $\overrightarrow{Q}$, or, equivalently, sequences with unit Hamming distance.[1] In particular, the non-zero entries of $\overrightarrow{Q}$ are determined by a shared $|\chi| \times |\chi|$ local rate matrix $\overrightarrow{Q}^{\text{tok}}$. Specifically, if two sequences $a$ and $b$ differ only at the $i$-th component, then their transition rate is given by $\overrightarrow{Q}_t(a, b) = \overrightarrow{Q}_t^{\text{tok}}(a_i, b_i)$.

Expressing the MI as the KL divergence between the joint distribution and the product of marginals, $\text{KL}\left[p_{XY} \parallel m_{XY}\right]$, requires training two separate score models, each tailored to a specific distribution. A carefully chosen transition matrix can circumvent this requirement, allowing for a single model to be trained instead. In particular, selecting $\overrightarrow{Q}_t^{\text{tok}} = \sigma(t)\overrightarrow{Q}_{\text{absorb}}^{\text{tok}}$, with $\sigma(t)$ a fixed scalar function, and the absorbing matrix $\overrightarrow{Q}_{\text{absorb}}^{\text{tok}}$ as defined in Lou et al. (2024); Campbell et al. (2022); Austin et al. (2021), ensures that the subcomponents can only transition into an absorbing state $\emptyset$. This choice is crucial because it enables the computation of marginal scores using a model trained solely on the joint distribution:

$$\frac{\overrightarrow{p}_t(\overrightarrow{X}_t = x, \overrightarrow{Y}_t = \emptyset)}{\overrightarrow{p}_t(\overrightarrow{X}_t = x', \overrightarrow{Y}_t = \emptyset)} = \frac{\overrightarrow{p}_t^X(x)}{\overrightarrow{p}_t^X(x')}, \qquad \frac{\overrightarrow{p}_t(\overrightarrow{X}_t = \emptyset, \overrightarrow{Y}_t = y)}{\overrightarrow{p}_t(\overrightarrow{X}_t = \emptyset, \overrightarrow{Y}_t = y')} = \frac{\overrightarrow{p}_t^Y(y)}{\overrightarrow{p}_t^Y(y')}. \tag{6}$$

---

[1]This prohibits any sequence to jump to another sequence with Hamming distance larger than one, but only in the infinitesimal time regime $dt$. Once we consider longer time intervals, perturbations immediately become non-local. Empirical evidence in the field of discrete diffusion models indicates that such decomposition allows dealing with complex distributions with non-local interactions, such as text/language and single-cell RNA sequencing data (Sahoo et al., 2024; Lou et al., 2024)

For the mathematical proof the reader is invited to check Appendix A.3. This result implies that a single score model trained on the joint distribution is sufficient for computing the marginal scores as well.[2] By integrating these design choices, we present the pseudocode for INFO-SEDD-J and INFO-SEDD-C in Appendix B, in Algorithm 1 and Algorithm 2 respectively.

**Theoretical properties.** We investigate the theoretical properties of INFO-SEDD, specifically focusing on the error decomposition and consistency. We establish that under mild assumptions on the boundedness of the score functions (constants $C_1, C_2$) and the neural network approximation errors ($\epsilon_p, \epsilon_q$), the deviation of the INFO-SEDD estimator from the true KL divergence is bounded by:

$$\left| \mathbb{E}_{x \sim p} \mathcal{E}(s_\theta^p, s_\phi^q; x) - \text{KL}\left[p \parallel q\right] \right| \leq \underbrace{\bar{\sigma}(T) D |\chi| \left(1 + \frac{C_2}{C_1}\right)(\epsilon_p + \epsilon_q)}_{\text{Estimation Error}} + \underbrace{(1 - \vec{p}_T(\emptyset^D))(DC_2 \log |\chi|)}_{\text{Truncation Bias}}.$$

$$(7)$$

Where $\mathcal{E}(s_\theta^p, s_\phi^q; x)$ corresponds to the right-hand side of Equation (5). This bound highlights a key trade-off: the first term, representing the estimation error, scales linearly with the score error and constants dependent on the data distribution. The second term represents a truncation bias arising from the finite time horizon $T$, which vanishes exponentially as the probability of the absorbing state $\vec{p}_T(\emptyset^D)$ approaches 1. Consequently, INFO-SEDD is a consistent estimator up to this exponentially decaying bias, allowing for accurate KL estimation without the exponential variance associated with standard importance sampling methods. We refer the reader to Appendix E for the full derivation and proofs.

**Estimating entropy.** Our method can also be used for Entropy estimation, since it can be expressed in terms of the KL divergence between a distribution $\vec{p}_0$ and the uniform distribution $\vec{u}_0$, as follows: $H(\vec{p}_0) = \log N - \text{KL}\left[\vec{p}_0 \parallel \vec{u}_0\right]$. Since the ratio $\frac{\vec{u}_t(x)}{\vec{u}_t(\emptyset)} = \frac{1}{N(e^{\bar{\sigma}(t)} - 1)}$, with $\bar{\sigma}(t) = \int_0^t \sigma(s) ds$ (see appendix A.2), we can adapt the formulation in Equation (5) to derive INFO-SEDD-H, an Entropy estimator, which is detailed in Appendix B, in Algorithm 3.

## 4 EXPERIMENTS

Next, we evaluate INFO-SEDD on a series of synthetic experiments, as well as on several realistic applications. Overall, while our competitors struggle with high MI scenarios and high-dimensional data, INFO-SEDD obtains accurate estimates. Using text and DNA data, we show that INFO-SEDD is not only accurate, but also useful for various downstream tasks. In Appendix D, we provide additional results, showing that INFO-SEDD can produce accurate Entropy estimates in Ising models (Onsager, 1944), which are used to understand the state of a particle lattice using their spins.

### 4.1 SYNTHETIC EXPERIMENTS

We validate our method on synthetic distributions with known mutual information (full details are in Appendix C.1). Given two vectors $X = (x_1, x_2, \ldots, x_D)$ and $Y = (y_1, y_2, \ldots, y_D)$ sampled from their respective discrete distributions, we denote vector dimensionality by $D$, and the support $|\chi|$ as the number of discrete values each element $x_i, y_i$ can take.

Next, we benchmark INFO-SEDD and several competitors on a high dimensional synthetic benchmark, by increasing both MI and $D$ as shown in Table 1. We compare the results of our proposed methodology against some of the estimators from Letizia et al. (2024), who introduce data derangements to improve bias and variances of existing variational estimators, including NWJ (Nguyen et al., 2007), SMILE (Song and Ermon, 2019a) and MINE (Belghazi et al., 2018). Letizia et al. (2024) also proposes the F-DIME estimators GAN-DIME, KL-DIME, HD-DIME, which are shown to work better at high MI. We also benchmark MINDE (Franzese et al., 2024), a generative neural estimator for continuous data. We use the same backbone for all methods (see Appendix C.1), with only minor tweaks to initial and final layers to accommodate the specifics of each method. For our method, we

---

[2]This configuration adds an absorbing state, increasing the dimension of the support

report the results with the INFO-SEDD-J variant, trained using the DWDSE loss. We train all methods using $10^5$ samples and a batch size of 1024 for $10^5$ steps.

Table 1: Results for the high dimensional synthetic benchmark for all estimators with given MI and same vector length (D) for each modality. We report the mean estimate and standard deviation over 10 seeds. Best estimates are marked in bold.

| Estimator | INFO-SEDD | GAN-DIME | HD-DIME | KL-DIME | MINDE | MINE | NWJ | SMILE |
|---|---|---|---|---|---|---|---|---|
| MI=10, D=10 | **9.92 ± 0.12** | 12.15 ± 0.89 | 9.73 ± 0.43 | 8.38 ± 0.90 | 14.01 ± 2.91 | 10.21 ± 6.33 | 6.16 ± 2.11 | 12.83 ± 0.95 |
| MI=20, D=20 | **20.02 ± 0.21** | 22.09 ± 1.75 | 12.65 ± 1.07 | 7.51 ± 0.56 | 26.98 ± 3.16 | 8.82 ± 0.80 | 6.50 ± 0.84 | 23.11 ± 1.41 |
| MI=30, D=30 | **29.83 ± 0.54** | 20.74 ± 1.75 | 11.72 ± 2.69 | 7.02 ± 0.43 | 31.08 ± 4.33 | 7.41 ± 1.23 | 6.35 ± 0.34 | 21.79 ± 1.08 |
| MI=40, D=40 | **39.11 ± 0.65** | 19.64 ± 1.33 | 11.68 ± 0.94 | 6.52 ± 0.32 | 33.97 ± 3.32 | 6.91 ± 0.66 | 6.24 ± 0.59 | 20.13 ± 1.27 |
| MI=50, D=50 | **47.77 ± 1.18** | 17.27 ± 1.46 | 10.47 ± 1.12 | 6.41 ± 0.62 | 32.60 ± 3.93 | 7.21 ± 1.14 | 5.95 ± 0.31 | 18.97 ± 1.05 |

Results in Table 1 demonstrate that INFO-SEDD consistently outperforms competing methods. INFO-SEDD performs better with high dimensional data and high MI, while existing neural estimators fail. This is not the only scenario where INFO-SEDD shines, as demonstrated extensively in Appendix C.1. In Appendix C.1.6 we perform an ablation study on $|\chi|$, showing that INFO-SEDD is robust to changes in support size, where, instead, the competitors cannot provide accurate estimates. Accuracy, however, is not the only dimension where competitors struggle. In Appendix C.1.3, we show that even when competing estimators like GAN-DIME and SMILE provide correct estimates, they take more epochs to converge compared to INFO-SEDD, resulting in slower training. Finally, in Appendix C.1.5 we evaluate empirically the sample complexity of INFO-SEDD observing that it is accurate even when using only $10^3$ samples.

Overall, our complete set of synthetic experiments confirms that INFO-SEDD is robust, efficient, and parsimonious in terms of sample size. Next, we revert to realistic use cases to further evaluate information-theoretic estimators in even more challenging scenarios.

## 4.2 APPLICATION: TEXT SUMMARIZATION

We consider the task of text summarization, which aims at generating concise document summaries, either by directly extracting sentences from the original corpus, or by introducing new, relevant phrases not present in the original text (Nallapati et al., 2016). We validate INFO-SEDD using the SUMMEVAL dataset (Fabbri et al., 2021) which assembles collections of summaries generated by 23 different models, providing human judgments for outputs generated by 15 of these models. We generate a collection of datasets of texts/summaries pairs from the SUMMEVAL dataset, such that we can scramble part of such pairs to gauge MI between texts and summaries distributions. We compare INFO-SEDD and alternative methods, showing how much each estimator is *consistent* with a "theoretical" trend of MI. We also study the application of INFO-SEDD for summarization model selection, by estimating the MI of each collection of model outputs with respect to the reference texts. Studying how these estimates compare to the human evaluations of the summaries helps practitioners to identify which human metric is best aligned to MI. For more details, refer to Appendix C.2.

**Consistency test.** We perform a consistency test for MI estimation between model-generated summaries and reference texts: we select the summaries generated by BART (Lewis et al., 2020) and pair them with the original text with a probability $\rho$ and with a random text in the dataset with probability $1 - \rho$. While we cannot establish an exact ground truth for these experiments, we can determine that MI should grow linearly as a function of $\rho$, under the assumption of MI being significantly larger than $\log 2$. This assumption is backed by studies (Takahira et al., 2016; Cover and King, 1978) that estimate the entropy rate of textual data to be larger than 1 bit per character (bpc), and by works that estimate MI in a similar context (Darrin et al., 2024). To support our claims and provide an order-of-magnitude estimate, we multiply the entropy rates of Takahira et al. (2016) and Cover and King (1978) with the average length of dataset summaries, to obtain entropy estimates for English texts with the same length, obtaining values of 256 nats and 303 nats, respectively. For a more detailed discussion, refer to Appendix C.2. We use the MDLM-SMALL model (Sahoo et al., 2024) as the backbone, with minimal changes to the architecture to accommodate our competitors. We also slightly modify the training strategy of Sahoo et al. (2024) to allow a more efficient learning of marginal and conditional scores for INFO-SEDD (check Appendix C.2 for details). This allows us to keep the same context length, architecture, and a similar number of parameters for all methods. Note that for all

competitors, we project text tokens into an embedding space of fixed dimension, by jointly learning an embedding look-up table for each method. For more experimental details, check Appendix C.2.

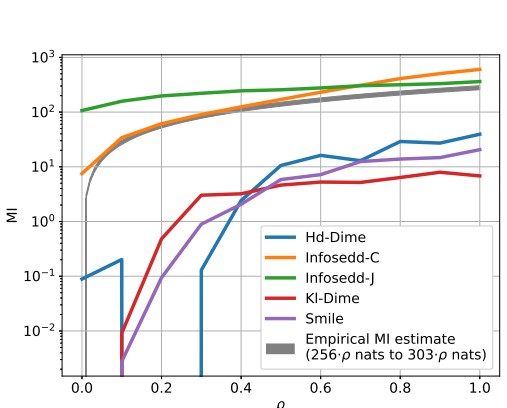

Figure 1: Consistency results for selected estimators.

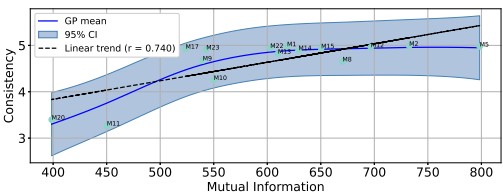

Figure 2: Consistency human metric compared to MI as estimated by INFO-SEDD-C.

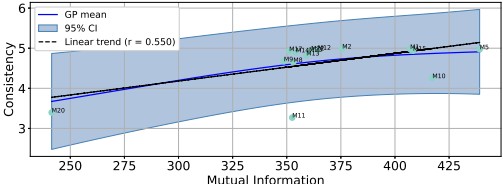

Figure 3: Consistency human metric compared to MI as estimated by INFO-SEDD-J.

As shown in Figure 1, where we report MI estimates for $\rho = 0.0, 0.1, ..., 1.0$, the consistency of both INFO-SEDD variants is the highest among all competitors, closely matching the empirical derivation outlined above. Note that INFO-SEDD-C obtains MI estimates closer to zero than the joint variant, when $\rho = 0.0$. Instead, variational approaches cannot reliably estimate MI with values larger than the logarithm of the batch size used for estimating it (McAllester and Stratos, 2020; Song and Ermon, 2019a). This results in MI being underestimated, a problem that could only be addressed by impractical batch sizes. Despite KL-DIME and SMILE exhibit an approximately linear correlation, they obtain low MI values, incompatible with the empirical derivation. HD-DIME can achieve higher values of MI, but fails for $\rho < 0.5$. MINDE does not provide meaningful MI estimates: this is likely due to the high embedding dimensionality which, together with the sequence length, is a challenging scenario for continuous score models.

**Model selection.** We now evaluate if MI is a meaningful signal to perform model selection, by measuring if MI estimates are compatible with human-preference metrics, as done by Darrin et al. (2024). This is done by estimating the MI of the text-model summary datasets in SUMMEVAL (Fabbri et al., 2021), for every model with available human metrics. The human metrics we consider are: **coherence**, which measures how well-structured and coherent the summary is; **consistency**, which quantifies how much the statements in the summary are entailed by the source; **fluency**, which evaluates the grammatical correctness and the formatting of the summary; and **relevance**, which measures if the summary captures only the most important information of the source text. Table 2, which shows Pearson correlation and Kendall's Tau test results, indicates that MI correlates the most with consistency. This is expected, as consistency effectively quantifies how much information the text and summary share. Note, however, that raters were asked to penalize hallucinated facts present in the summaries, which is not captured by MI. The second metric which correlates the most with MI is fluency. The definition of fluency does not take into account a comparison between summary and source text, but it also strongly correlates with consistency. Thirdly, MI correlates well with relevance. Relevance considers a comparison between summary and source text in terms of what is considered important information by human annotators. However, important information is highly subjective, and human intentions are likely not captured by MI. Lastly, coherence does not quantify a relationship between source text and summaries, which explains the low correlation with MI. In Figures 2 and 3 we show the relation between the MI estimates of INFO-SEDD-C and INFO-SEDD-J and consistency by running a Gaussian Process (GP) regression with a Matérn Kernel with smoothing parameter $\nu = 1.5$ and noise level $0.01$. We report the GP mean, its 95% confidence interval and the Pearson correlation. Figures 2 and 3 show that consistency saturates around the maximum score, whereas MI does not have the same ceiling effect. The reader is referred to Appendix C.2 for a similar discussion about the other metrics. Overall, our results are in line with

the findings by Darrin et al. (2024), where INFO-SEDD-C achieves a comparable correlation with the most important metric, consistency, albeit without requiring elaborate embedding models, that rely on a large set of parameters to be learned.

Table 2: Pearson (left) and Kendall's Tau (right) correlation of human metrics and MI for INFO-SEDD and selected competitors. The labels are shorthand notations for the human metrics: COH=coherence, CON=consistency, FLU=fluency, REL=relevance, OVR=overall. The reader is invited to check Appendix C.3 for correlations with other metrics offered by the SUMMEVAL framework.

| | COH | CON | FLU | REL | OVR | | COH | CON | FLU | REL | OVR |
|---|---|---|---|---|---|---|---|---|---|---|---|
| INFO-SEDD-C | 0.209 | 0.740 | 0.679 | 0.411 | 0.568 | INFO-SEDD-C | 0.105 | 0.505 | 0.134 | 0.200 | 0.219 |
| INFO-SEDD-J | -0.091 | 0.550 | 0.455 | 0.288 | 0.322 | INFO-SEDD-J | 0.048 | 0.486 | 0.153 | 0.219 | 0.238 |
| KL-DIME | 0.170 | 0.214 | 0.194 | 0.076 | 0.193 | KL-DIME | 0.105 | 0.429 | 0.096 | 0.048 | 0.067 |
| HD-DIME | -0.243 | 0.331 | 0.281 | -0.145 | 0.063 | HD-DIME | -0.162 | -0.029 | 0.057 | -0.067 | -0.048 |
| SMILE | -0.367 | -0.074 | -0.162 | -0.149 | -0.221 | SMILE | -0.238 | -0.105 | 0.019 | -0.067 | -0.086 |

## 4.3 APPLICATION: GENOMICS

The genomics field heavily relies on computational methods to improve our understanding of hidden patterns in large and complex genomic data sets from basic and clinical research projects (Libbrecht and Noble, 2015; Whalen et al., 2022; Teschendorff and Horvath, 2025). We consider two fundamental problems in genetics (genome classification and motif identification in genetic promoters) that rely on DNA sequencing data: DNA is a nucleotide made of four types of nitrogen bases, Adenine (A), Thymine (T), Guanine (G), and Cytosine (C). The order, or sequence, of these bases determines the biological instructions that are contained in a strand of DNA. We follow the recent practice of considering DNA sequencing data as text (Dotan et al., 2024; Qiao et al., 2024; Malusare et al., 2024; Eapen, 2025), albeit we adopt the simple tokenization technique of considering each DNA base in a sequence as a token. This results in DNA sequences being modeled as high-dimensional vectors with a support of four elements per sequence element ($|\chi| = 4$). Additional details are in Appendix C.4.

**Consistency test.** We assess the reliability of INFO-SEDD in a low MI regime, and consider the "HUMAN VS. WORM" dataset, which is part of the Genomics Benchmarks by Grešová et al. (2023), a recently proposed suite with eight regulatory element classification tasks. The dataset consists of $10^5$ DNA sequences, paired with 2 class labels (human or worm), with a median length of 200 bases. We perform a consistency test as follows: for each sample in the dataset, we randomize the label with probability $\rho$ and compute the MI between the DNA sequences and the new set of labels. To construct a reference, we can approximate order-of-magnitude and slope of ground-truth MI values with the assumption that, given a classifier trained to predict the label given the genome, the classification error is approximately the same across each sample. To do so, we express MI as $I(X, Y) = H(Y) - H(Y|X)$, where $Y$ is the label and $X$ is the genome, and approximate $H(Y|X)$ as $H_b(\text{Acc.})$, where $H_b$ is the binary entropy and Acc. is the accuracy of the classifier. We estimate $H(Y)$ directly from data using a bin estimate given that we have a binary label and enough samples. In our experiments, we use a pretrained CADUCEUS model (Schiff et al., 2024) for the backbone of all methods, with minimal architectural changes when needed, ensuring a fair comparison. For more details check Appendix C.4.

Results in Figure 4 show that INFO-SEDD-C outperforms the competitors, as it closely matches the *classifier-based* reference MI. HD-DIME performs relatively well for $\rho < 1.0$ while other competitors struggle to keep increasingly high MI estimates for growing values of $\rho$. The difference in performance between the two INFO-SEDD variants is due to the difference in dimensionality between the label, mono-dimensional, and the DNA sequence. This makes the optimization process of INFO-SEDD-C significantly easier, since it only requires parametrizing the scores of the marginal distribution of the label and of the conditional distribution of the label given the associated DNA sequence. INFO-SEDD-J, instead, must parametrize the score of the joint distribution of the DNA sequence and label, a considerably more difficult task since it needs to do discrete diffusion training on the DNA sequence while INFO-SEDD-C only treats the DNA sequence as a conditioning signal.

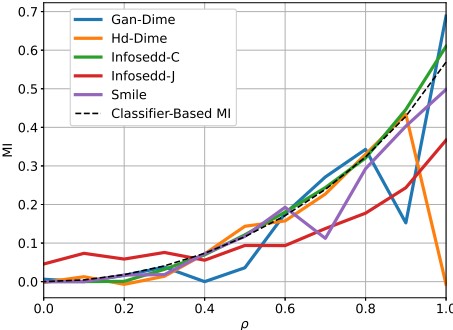

Figure 4: Consistency test for the HUMAN VS. WORM dataset against the best performing competitors. For the complete numerical results check Appendix C.4.

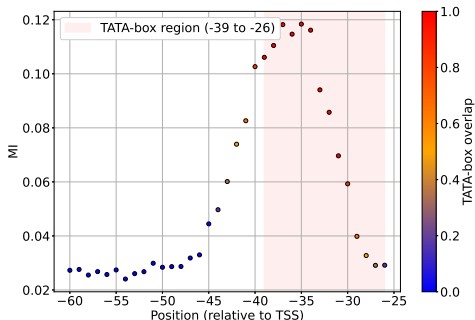

Figure 5: *Arabidopsis thaliana* promoter TATA-BOX identification on DNA sequences using INFO-SEDD.

**Finding motifs in promoters.** A central issue in molecular biology is understanding the regulatory mechanisms that control gene expression. In particular, gene expression requires a stretch of regulatory DNA called a promoter which contains certain motifs, i.e., patterns that show a statistically significant dependency with expression levels. Computational methods based on MI (Elemento et al., 2007; Rao et al., 2007) use whole genome sequences to search for the key elements in transcription regulation. The key idea is to directly quantify the dependency between the presence or absence of a given motif in a regulatory region and the expression of the corresponding gene.

Umarov and Solovyev (2017) propose a convolutional classifier for building eukaryotic promoter recognition models, which they evaluate on numerous promoter sequences extracted from the well-known EPD database (Dreos et al., 2013). In this work, we focus on *Arabidopsis thaliana* TATA-promoter/non-promoter sequences, consisting of 1497 samples and 2879 samples respectively, each of fixed length of 251 DNA bases. A minimal eukaryotic promoter region contains a transcription start site (TSS) and a so-called TATA-BOX motif at a position $\approx 30$ base pair upstream from the transcription start site, which is a key feature used for classification. The *Arabidopsis thaliana* is no stranger to this rule, as the preferred position for the TATA-BOX motif is between -39 and -26 with respect to TSS (Bernard et al., 2010). Umarov and Solovyev (2017) propose a randomized technique, which we extend to our method, to find the TATA-BOX motif: to discover such sites we train INFO-SEDD such that it can estimate the MI between the sequence distribution and the promoter labels. Then, we use a sliding window of length $L$ and mask the DNA sequence outside this window. Using sliding windows moving from the beginning of a functional site sequence, we can build a MI profile that reflects the effect of a random sequence. Whenever the window covers a segment that is irrelevant for determining whether a sequence is a promoter or not, MI remains low. Instead, unmasking the TATA-BOX motif implies a substantial increase in MI values. Differently from the method in (Umarov and Solovyev, 2017), our approach is robust to correlated motifs: masking only one correlated motif results in high classification accuracy due to the other motif still being unmasked; our method, instead, only unmasks the current motif, which allows assessing its importance without interference from correlated motifs. For MI estimation we fine-tune a pretrained CADUCEUS model, used as the backbone for INFO-SEDD-J. We use the joint variant since its training procedure enables label prediction with DNA sequence masking, while the other variant only models the conditional distribution given unmasked data (more details in Appendix C.4).

Figure 5 shows the MI profile between the window and the promoter label: red dots imply a high overlap between the window and the TATA-BOX location, as determined by Bernard et al. (2010). While known from the genomics literature, INFO-SEDD can effectively locate the TATA-BOX using MI, which makes it an invaluable tool for motif discovery. Note also that other MI estimators would need different training runs for each window, whereas INFO-SEDD natively supports MI estimation between subsets of DNA sequences. This property unlocks applications of MI which were previously hindered by inaccurate estimators and poorly scalable methods.

## 5 CONCLUSION

The ability to measure the non-linear relation among arbitrary data distributions stands as a cornerstone for scientific discovery, but scalable, and efficient methods to accurately estimate information theoretic metrics, without the need for pre-processing workarounds, has only recently attracted the attention of the community. In this work, we contributed a novel method tailored to estimate such metrics for a particularly challenging case, that of high-dimensional discrete data distributions. To the best of our knowledge, our method is unique, as it is scalable, consistent, and accurate, as demonstrated by our experimental validation; moreover, our approach is efficient, as it can leverage existing pretrained models and does not require ad-hoc procedures to work with discrete data. Remarkably INFO-SEDD remains flexible to the choice of masked discrete diffusion model, enabling the usage of other models and parameterizations such as MD4 (Shi et al., 2024) or LLADA (Nie et al., 2026).

Armed with INFO-SEDD, we demonstrated that the ability to estimate MI between discrete random variables is key for a number of applications. In the context of text summarization, MI is a meaningful proxy to judge performance, and it can be used for model selection. In the genetics domain, we showed that access to accurate MI estimates has the potential of enabling new discoveries using sequencing data, which is discrete by nature. Moreover, the solid mathematical foundations of INFO-SEDD enable exciting extensions to mixed continuous/discrete data using the Generator Matching framework (Holderrieth et al., 2025), with many potential applications in diverse scientific fields.

## REPRODUCIBILITY STATEMENT

We provide the code at `https://github.com/AlbertoForesti/mutinfo-diffusion`, where we also provide instructions and utility scripts to build datasets and run experiments. In Appendix C.2.2 we report the model setup for text experiments and in Appendix C.4.2. Likewise, details and assumptions made in building the datasets are provided in Appendix C.1.1 for synthetic dataset, Appendix C.2.1 for text and Appendix C.4.1 for DNA. We do a performance analysis of the benchmarked methods using the same codebase and the same machine, with the only differences being the methodology in MI estimation and loss functions.

## ACKNOWLEDGEMENTS

Alberto Foresti, Giulio Franzese and Pietro Michiardi were partially funded by project MUSECOM2- AI-enabled MUltimodal SEmantic COMmunications and COMputing, in the Machine Learning based Communication Systems, towards Wireless AI (WAI), Call 2022, ChistERA.

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

# Information Estimation with Discrete Diffusion Supplementary Material

# A MATHEMATICAL PROOFS OF THE MAIN METHOD

## A.1 PROOF OF EQUATION 4

We first break $\mathbb{E}\left[\log \frac{\overleftarrow{p}_T}{\overleftarrow{q}_T}\Big|\overleftarrow{X}_0\right]$ in $\mathbb{E}\left[\log \overleftarrow{p}_T\Big|\overleftarrow{X}_0\right] - \mathbb{E}\left[\log \overleftarrow{q}_T\Big|\overleftarrow{X}_0\right]$ and apply the Dynkin's formula separately. We start with $\mathbb{E}\left[\log \overleftarrow{p}_T\Big|\overleftarrow{X}_0\right]$:

$$\mathbb{E}\left[\log \overleftarrow{p}_T\Big|\overleftarrow{X}_0\right] = \log \overleftarrow{p}_0(\overleftarrow{X}_0) + \mathbb{E}\left[\int_0^T \frac{\partial \log \overleftarrow{p}_t}{\partial t}(\overleftarrow{X}_t, t) + \mathcal{B}[\log \overleftarrow{p}_t](\overleftarrow{X}_t, t)dt\Big|\overleftarrow{X}_0\right]$$

$$= \log \overleftarrow{p}_0(\overleftarrow{X}_0) + \mathbb{E}\left[\int_0^T \frac{\partial \log \overleftarrow{p}_t}{\partial t}(\overleftarrow{X}_t, t) + \sum_{x \neq \overleftarrow{X}_t} \overleftarrow{Q}_t(x, \overleftarrow{X}_t)(\log \overleftarrow{p}_t(x) - \log \overleftarrow{p}_t(\overleftarrow{X}_t))dt\Big|\overleftarrow{X}_0\right]$$

$$= \log \overleftarrow{p}_0(\overleftarrow{X}_0) + \mathbb{E}\left[\int_0^T \frac{\partial \log \overleftarrow{p}_t}{\partial t}(\overleftarrow{X}_t, t) + \sum_{x \neq \overleftarrow{X}_t} \overleftarrow{Q}_t(x, \overleftarrow{X}_t)\log \frac{\overleftarrow{p}_t(x)}{\overleftarrow{p}_t(\overleftarrow{X}_t)}dt\Big|\overleftarrow{X}_0\right]$$

We now focus on the term $\frac{\partial \log \overleftarrow{p}_t}{\partial t}$, which we can rewrite using $\partial \overleftarrow{p}_t = \overleftarrow{Q}_t \overleftarrow{p}_t$:

$$\frac{\partial \log \overleftarrow{p}_t}{\partial t} = \partial \overleftarrow{p}_t \Big/ \overleftarrow{p}_t = \frac{\overleftarrow{Q}_t \overleftarrow{p}_t}{\overleftarrow{p}_t}$$

Where the division between numerator and denominator in the last two terms denotes element-wise division. We focus now on simplifying the numerator, using the definition in equation 1. Recalling that

$$\overleftarrow{Q}_t(b, a) = \left(\frac{\overrightarrow{p}_{T-t}(b)}{\overrightarrow{p}_{T-t}(a)}\overrightarrow{Q}_{T-t}(a, b)\right)(1 - \delta(a, b)) + \left(-\sum_{b \neq a}\overleftarrow{Q}_t(b, a)\right)\delta(a, a)$$

$$= \left(\frac{\overleftarrow{p}_t(b)}{\overleftarrow{p}_t(b)}\overrightarrow{Q}_{T-t}(a, b)\right)(1 - \delta(a, b)) + \left(-\sum_{b \neq a}\overleftarrow{Q}_t(b, a)\right)\delta(a, a)$$

we compute the $a$-th element of $\overleftarrow{Q}_t \overleftarrow{p}_t$:

$$[\overleftarrow{Q}_t \overleftarrow{p}_t](a) = \sum_b \overleftarrow{Q}_t(a, b)\overleftarrow{p}_t(b)$$

$$= \sum_{b \neq a} \overleftarrow{Q}_t(a, b)\overleftarrow{p}_t(b) + \overleftarrow{Q}_t(a, a)\overleftarrow{p}_t(a)$$

$$= \sum_{b \neq a} \overrightarrow{Q}_{T-t}(b, a)\frac{\overleftarrow{p}_t(a)}{\overleftarrow{p}_t(b)}\overleftarrow{p}_t(b) - \sum_{b \neq a}\overleftarrow{Q}_t(b, a)\overleftarrow{p}_t(a)$$

$$= \sum_{b \neq a} \overrightarrow{Q}_{T-t}(b, a)\overleftarrow{p}_t(a) - \sum_{b \neq a}\overrightarrow{Q}_{T-t}(a, b)\frac{\overleftarrow{p}_t(b)}{\overleftarrow{p}_t(a)}\overleftarrow{p}_t(a)$$

$$= \sum_{b \neq a} \overrightarrow{Q}_{T-t}(b, a)\overleftarrow{p}_t(a) - \sum_{b \neq a}\overrightarrow{Q}_{T-t}(a, b)\overleftarrow{p}_t(b)$$

$$= \sum_{b \neq a} \overrightarrow{Q}_{T-t}(b, a)\overleftarrow{p}_t(a) - \overrightarrow{Q}_{T-t}(a, b)\overleftarrow{p}_t(b)$$

Finally, if we divide by the denominator we obtain:

$$
\left[ \frac{\overleftarrow{Q}_t \overleftarrow{p}_t}{\overleftarrow{p}_t} \right](a) = \frac{\sum_{b \neq a} \overrightarrow{Q}_{T-t}(b,a) \overleftarrow{p}_t(a) - \overrightarrow{Q}_{T-t}(a,b) \overleftarrow{p}_t(b)}{\overleftarrow{p}_t(a)}
$$
$$
= \sum_{b \neq a} \overrightarrow{Q}_{T-t}(b,a) - \overrightarrow{Q}_{T-t}(a,b) \frac{\overleftarrow{p}_t(b)}{\overleftarrow{p}_t(a)}
$$

Moreover, if we notice that $\overleftarrow{Q}_t(x, \overleftarrow{X}_t) \log \frac{\overleftarrow{p}_t(x)}{\overleftarrow{p}_t(\overleftarrow{X}_t)} = \overrightarrow{Q}_{T-t}(\overleftarrow{X}_t, x) \frac{\overleftarrow{p}_t(x)}{\overleftarrow{p}_t(\overleftarrow{X}_t)} \log \frac{\overleftarrow{p}_t(x)}{\overleftarrow{p}_t(\overleftarrow{X}_t)}$, we can write the Dynkin's formula as:

$$
\mathbb{E}\left[ \log \overleftarrow{p}_T | \overleftarrow{X}_0 \right] =
$$
$$
\log \overleftarrow{p}_0(\overleftarrow{X}_0) +
$$
$$
\mathbb{E}\left[ \int_0^T \sum_{x \neq \overleftarrow{X}_t} \overrightarrow{Q}_{T-t}(x, \overleftarrow{X}_t) - \overrightarrow{Q}_{T-t}(\overleftarrow{X}_t, x) \frac{\overleftarrow{p}_t(x)}{\overleftarrow{p}_t(\overleftarrow{X}_t)} + \overrightarrow{Q}_{T-t}(\overleftarrow{X}_t, x) \frac{\overleftarrow{p}_t(x)}{\overleftarrow{p}_t(\overleftarrow{X}_t)} \log \frac{\overleftarrow{p}_t(x)}{\overleftarrow{p}_t(\overleftarrow{X}_t)} dt \Big| \overleftarrow{X}_0 \right]
$$
$$(8)$$

Then, if we define $K(\alpha) = \alpha(\log \alpha - 1)$ and group some terms we obtain:

$$
\mathbb{E}\left[ \log \overleftarrow{p}_T | \overleftarrow{X}_0 \right] =
$$
$$
\log \overleftarrow{p}_0(\overleftarrow{X}_0) + \mathbb{E}\left[ \int_0^T \sum_{x \neq \overleftarrow{X}_t} \overrightarrow{Q}_{T-t}(x, \overleftarrow{X}_t) + \overrightarrow{Q}_{T-t}(\overleftarrow{X}_t, x) K\left( \frac{\overleftarrow{p}_t(x)}{\overleftarrow{p}_t(\overleftarrow{X}_t)} \right) dt \Big| \overleftarrow{X}_0 \right]
$$

We now repeat similar calculations for $\mathbb{E}\left[ \log \overleftarrow{q}_T \Big| \overleftarrow{X}_0 \right]$. Firstly, the term $\frac{\partial \log \overleftarrow{q}_t}{\partial t}$ becomes:

$$
\left[ \frac{\partial \log \overleftarrow{q}_t}{\partial t} \right](a) = \sum_{b \neq a} \overrightarrow{Q}_{T-t}(b,a) - \overrightarrow{Q}_{T-t}(a,b) \frac{\overleftarrow{q}_t(b)}{\overleftarrow{q}_t(a)}
$$

Whereas the backward operator term $\mathcal{B}[\log \overleftarrow{q}_t](\overleftarrow{X}_t, t)$ becomes:

$$
\overleftarrow{Q}_t(x, \overleftarrow{X}_t) \log \frac{\overleftarrow{q}_t(x)}{\overleftarrow{q}_t(\overleftarrow{X}_t)} = \overrightarrow{Q}_{T-t}(\overleftarrow{X}_t, x) \frac{\overleftarrow{p}_t(x)}{\overleftarrow{p}_t(\overleftarrow{X}_t)} \log \frac{\overleftarrow{q}_t(x)}{\overleftarrow{q}_t(\overleftarrow{X}_t)}
$$

Combining everything together gives:

$$
\mathbb{E}\left[ \log \overleftarrow{q}_T \Big| \overleftarrow{X}_0 \right] =
$$
$$
\log \overleftarrow{q}_0(\overleftarrow{X}_0) +
$$
$$
\mathbb{E}\left[ \int_0^T \sum_{x \neq \overleftarrow{X}_t} \overrightarrow{Q}_{T-t}(x, \overleftarrow{X}_t) - \overrightarrow{Q}_{T-t}(\overleftarrow{X}_t, x) \frac{\overleftarrow{q}_t(x)}{\overleftarrow{q}_t(\overleftarrow{X}_t)} + \overrightarrow{Q}_{T-t}(\overleftarrow{X}_t, x) \frac{\overleftarrow{p}_t(x)}{\overleftarrow{p}_t(\overleftarrow{X}_t)} \log \frac{\overleftarrow{q}_t(x)}{\overleftarrow{q}_t(\overleftarrow{X}_t)} dt \Big| \overleftarrow{X}_0 \right]
$$
$$(9)$$

Finally, we can estimate $\mathbb{E}\left[\log\frac{\overleftarrow{p}_T}{\overleftarrow{q}_T}\Big|\overleftarrow{X}_0\right]$ by subtracting equation 9 from equation 8:

$$\mathbb{E}\left[\log\frac{\overleftarrow{p}_T}{\overleftarrow{q}_T}\Big|\overleftarrow{X}_0\right] \approx \mathbb{E}\left[\int_0^T \sum_{x\neq\overleftarrow{X}_t}\overrightarrow{Q}_{T-t}(\overleftarrow{X}_t,x)\left(K\left(\frac{\overleftarrow{p}_t(x)}{\overleftarrow{p}_t(\overleftarrow{X}_t)}\right)+\frac{\overleftarrow{q}_t(x)}{\overleftarrow{q}_t(\overleftarrow{X}_t)}-\frac{\overleftarrow{p}_t(x)}{\overleftarrow{p}_t(\overleftarrow{X}_t)}\log\frac{\overleftarrow{q}_t(x)}{\overleftarrow{q}_t(\overleftarrow{X}_t)}\right)dt\Big|\overleftarrow{X}_0\right]$$

By using the fact that $\mathbb{E}\left[\mathbb{E}\left[\log\frac{\overleftarrow{p}_T}{\overleftarrow{q}_T}\Big|\overleftarrow{X}_0\right]\right]=\mathbb{E}\left[\log\frac{\overleftarrow{p}_T}{\overleftarrow{q}_T}\right]$, $\overleftarrow{p}_t=\overrightarrow{p}_{T-t}$, $\overleftarrow{q}_t=\overrightarrow{q}_{T-t}$, $\overleftarrow{X}_t=\overrightarrow{X}_{T-t}$ and by setting $\tau=T-t$, we get:

$$\mathbb{E}\left[\int_0^T \sum_{x\neq\overrightarrow{X}_\tau}\overrightarrow{Q}_\tau(\overrightarrow{X}_\tau,x)\left(K\left(\frac{\overrightarrow{p}_\tau(x)}{\overrightarrow{p}_\tau(\overrightarrow{X}_\tau)}\right)+\frac{\overrightarrow{q}_\tau(x)}{\overrightarrow{q}_\tau(\overrightarrow{X}_\tau)}-\frac{\overrightarrow{p}_\tau(x)}{\overrightarrow{p}_\tau(\overrightarrow{X}_\tau)}\log\frac{\overrightarrow{q}_\tau(x)}{\overrightarrow{q}_\tau(\overrightarrow{X}_\tau)}\right)d\tau\right]$$

recovering Equation (4).

## A.2 ENTROPY ESTIMATION DETAILS

We provide hereafter for completeness the steps necessary to show that $\frac{\overrightarrow{u}_t(x)}{\overrightarrow{u}_t(\emptyset)}=\frac{1}{N(e^{\overline{\sigma}(t)}-1)}$ for $x\neq\emptyset$:

$$\frac{\overrightarrow{u}_t(x)}{\overrightarrow{u}_t(\emptyset)}=\frac{\sum_{x_0\in\chi}\overrightarrow{u}_t(x|x_0)\overrightarrow{u}_0(x_0)}{\sum_{x_0\in\chi}\overrightarrow{u}_t(\emptyset|x_0)\overrightarrow{u}_0(x_0)}=\frac{\sum_{x_0\in\chi}\delta(x,x_0)e^{-\overline{\sigma}(t)}\frac{1}{N}}{\sum_{x_0\in\chi}(1-e^{-\overline{\sigma}(t)})\frac{1}{N}}=\frac{e^{-\overline{\sigma}(t)}}{N(1-e^{-\overline{\sigma}(t)})}=\frac{1}{N(e^{\overline{\sigma}(t)}-1)}$$

## A.3 PROOF OF EQUATION (6)

Consider $\overrightarrow{p}_t(\overrightarrow{X}_t=\bar{x},\overrightarrow{Y}_t=\emptyset)$:

$$\overrightarrow{p}_t(\overrightarrow{X}_t=\bar{x},\overrightarrow{Y}_t=\emptyset)=\sum_{x,y}\Pr(\overrightarrow{X}_t=\bar{x},\overrightarrow{Y}_t=\emptyset,\overrightarrow{X}_0=x,\overrightarrow{Y}_0=y)\tag{10}$$

$$=\sum_{x,y}\underbrace{\Pr(\overrightarrow{Y}_s=\emptyset\,|\,\overrightarrow{X}_t=\bar{x},\overrightarrow{X}_0=x,\overrightarrow{Y}_0=y)}_{\Pr(\overrightarrow{Y}_s\text{ jumps to }\emptyset\text{ in }[0,t])}\Pr(\overrightarrow{X}_t=\bar{x}\,|\,\overrightarrow{X}_0=x,\overrightarrow{Y}_0=y)\Pr(\overrightarrow{X}_0=x,\overrightarrow{Y}_0=y)$$

$$=\sum_{x,y}\Pr(\overrightarrow{Y}_s\text{ jumps to }\emptyset\text{ in }[0,t])\Pr(\overrightarrow{X}_t=\bar{x}\,|\,\overrightarrow{X}_0=x)\Pr(\overrightarrow{X}_0=x,\overrightarrow{Y}_0=y)$$

$$=\Pr(\overrightarrow{Y}_s\text{ jumps to }\emptyset\text{ in }[0,t])\sum_x\Pr(\overrightarrow{X}_t=\bar{x}\,|\,\overrightarrow{X}_0=x)\underbrace{\sum_y\Pr(\overrightarrow{X}_0=x,\overrightarrow{Y}_0=y)}_{\Pr(\overrightarrow{X}_0=x)}$$

$$=\Pr(\overrightarrow{Y}_s\text{ jumps to }\emptyset\text{ in }[0,t])\Pr(\overrightarrow{X}_t=\bar{x})$$

Equation (10) implies that $\frac{\overrightarrow{p}_t(\overrightarrow{X}_t=x,\overrightarrow{Y}_t=\emptyset)}{\overrightarrow{p}_t(\overrightarrow{X}_t=\bar{x},\overrightarrow{Y}_t=\emptyset)}=\frac{\Pr(\overrightarrow{X}_t=x)}{\Pr(\overrightarrow{X}_t=\bar{x})}$. This important property enables the estimation of mutual information without modifying the score network.

## B ALGORITHM PSEUDO-CODE

In this Section we show the pseudocodes for the different INFO-SEDD variants.

---

**Algorithm 1** INFO-SEDD-J: Estimate $I(\mathbf{x}, \mathbf{y})$

---

**Require:** Initial sample $[\vec{X}_0, \vec{Y}_0] \sim \vec{p}_0$, score network $s_\theta$
1: $t \sim u(0, T)$ {Sample time uniformly}
2: $[\vec{X}_t, \vec{Y}_t] \sim \vec{p}_t(\cdot|[\vec{X}_0, \vec{Y}_0])$ {Perturb data}
3: $\hat{I} = 0$
4: **for** $i : \vec{X}_t^i = \emptyset$ **do**
5:     **for** $n \in [1 : N]$ **do**
6:         $\tilde{X} = [\vec{X}_t^1, \ldots, \vec{X}_t^{i-1}, n, \vec{X}_t^{i+1}, \ldots, \vec{X}_t^M]$
7:         $\hat{I} + = T\sigma(t)\left(K\left(s_\theta([\vec{X}_t, \vec{Y}_t])_{[\tilde{X}, \vec{Y}_t]}\right) + s_\theta([\vec{X}_t, \emptyset])_{[\tilde{X}, \emptyset]} - s_\theta([\vec{X}_t, \vec{Y}_t])_{[\tilde{X}, \vec{Y}_t]}\log\left(s_\theta([\vec{X}_t, \emptyset])_{[\tilde{X}, \emptyset]}\right)\right)$
8:     **end for**
9: **end for**
10: **for** $i : \vec{Y}_t^i = \emptyset$ **do**
11:     **for** $n \in [1 : N]$ **do**
12:         $\tilde{Y} = [\vec{Y}_t^1, \ldots, \vec{Y}_t^{i-1}, n, \vec{Y}_t^{i+1}, \ldots, \vec{Y}_t^M]$
13:         $\hat{I} + = T\sigma(t)\left(K\left(s_\theta([\vec{X}_t, \vec{Y}_t])_{[\vec{X}_t, \tilde{Y}]}\right) + s_\theta([\emptyset, \vec{Y}_t])_{[\emptyset, \tilde{Y}]} - s_\theta([\vec{X}_t, \vec{Y}_t])_{[\vec{X}_t, \tilde{Y}]}\log\left(s_\theta([\emptyset, \vec{Y}_t])_{[\emptyset, \tilde{Y}]}\right)\right)$
14:     **end for**
15: **end for**
16: **return** $\hat{I}$

---

**Algorithm 2** INFO-SEDD-C: Estimate $I(\mathbf{x}, \mathbf{y})$

---

**Require:** Initial sample $[\vec{X}_0, \vec{Y}_0] \sim \vec{p}_0$, score network $s_\theta$
1: $t \sim u(0, T)$ {Sample time uniformly}
2: $\vec{Y}_t \sim \vec{p}_t(\cdot|\vec{Y}_0)$ {Perturb data}
3: $\hat{I} = 0$
4: **for** $i : \vec{Y}_t^i = \emptyset$ **do**
5:     **for** $n \in [1 : N]$ **do**
6:         $\tilde{X} = [\vec{Y}_t^1, \ldots, \vec{Y}_t^{i-1}, n, \vec{Y}_t^{i+1}, \ldots, \vec{Y}_t^M]$
7:         $\hat{I} + = T\sigma(t)\left(K\left(s_\theta([\vec{X}_0, \vec{Y}_t])_{[\vec{X}_0, \tilde{Y}]}\right) + s_\theta([\emptyset, \vec{Y}_t])_{[\emptyset, \tilde{Y}]} - s_\theta([\vec{X}_t, \vec{Y}_0])_{[\tilde{X}, \vec{Y}_0]}\log\left(s_\theta([\emptyset, \vec{Y}_t])_{[\emptyset, \tilde{Y}]}\right)\right)$
8:     **end for**
9: **end for**
10: **return** $\hat{I}$

---

**Algorithm 3** INFO-SEDD-H: Estimate $H(\mathbf{x})$

---

**Require:** Initial sample $\vec{X}_0 \sim \vec{p}_0$, score network $s_\theta$
1: $t \sim u(0, T)$ {Sample time uniformly}
2: $\vec{X}_t \sim \vec{p}_t(\cdot|\vec{X}_0)$ {Perturb data}
3: $\hat{H} = 0$
4: **for** $i : \vec{X}_t^i = \emptyset$ **do**
5:     **for** $n \in [1 : N]$ **do**
6:         $\tilde{X} = [\vec{X}_t^1, \ldots, \vec{X}_t^{i-1}, n, \vec{X}_t^{i+1}, \ldots, \vec{X}_t^M]$
7:         $\hat{H} + = T\sigma(t)\left(K\left(s_\theta(\vec{X}_t)_{\tilde{X}}\right) + \frac{1}{N(e^{\bar{\sigma}(t)} - 1)} - s_\theta(\vec{X}_t)_{\tilde{X}}\log\left(\frac{1}{N(e^{\bar{\sigma}(t)} - 1)}\right)\right)$
8:     **end for**
9: **end for**
10: **return** $\hat{H}$

---

## C EXPERIMENTAL DETAILS

### C.1 SYNTHETIC EXPERIMENTS

#### C.1.1 DATASET CONSTRUCTION

In this section, we show how we construct the datasets for our synthetic experiments. We generate joint distributions for random variables $X, Y$ with user-defined mutual information and support sizes $\chi_X, \chi_Y$. Using an evolutionary strategy, we encode the joint distribution in a vector $g_a \in \mathbb{R}^{|\chi_X||\chi_Y|}$ and transform it into a valid probability distribution via normalization and reshaping: $\frac{g_a - \min(g_a)\mathbf{1} + \epsilon\mathbf{1}}{\mathbf{1}(g_a - \min(g_a)\mathbf{1} + \epsilon\mathbf{1})}$ where $\epsilon$ ensures full support. The mutual information, computable in closed form, serves as the selection criterion in the evolutionary process. For large $|\chi_X|$ and $|\chi_Y|$, the evolutionary strategy struggles. Instead, we generate high mutual information distributions by concatenating independent distributions, leveraging the additive property of mutual information. Additionally, isomorphisms like Cantor's pairing function, $\pi(x, y) = \frac{1}{2}(x + y)(x + y + 1) + y$, enables support expansion without altering mutual information, aiding consistency across experiments.

More formally, we exploit the additivity of mutual information with independent random variables to generate complex datasets. By appending discrete noise random variables $Z_{\mathbf{x}}, Z_{\mathbf{y}}$ to the original random variables $\mathbf{x}, \mathbf{y}$, we have $I(\mathbf{x}, \mathbf{y}) = I([\mathbf{x}, Z_{\mathbf{x}}], [\mathbf{y}, Z_{\mathbf{y}}])$. Pairing functions are isomorphisms that map $\mathbb{N} \times \mathbb{N}$ to $\mathbb{N}$. They allow preserving mutual information through the Markov Chain:

$$[\mathbf{x}, Z_{\mathbf{x}}] \to \hat{\mathbf{x}} \to \hat{\mathbf{y}} \to [\mathbf{y}, Z_{\mathbf{y}}] \to \hat{\mathbf{y}} \to \mathbf{x} \to [\mathbf{x}, Z_{\mathbf{x}}]$$

Where, by the *data processing inequality*, we have $I([\mathbf{x}, Z_{\mathbf{x}}], [\mathbf{y}, Z_{\mathbf{y}}]) \geq I(\hat{\mathbf{x}}, \hat{\mathbf{y}}) \geq I([\mathbf{x}, Z_{\mathbf{x}}], [\mathbf{y}, Z_{\mathbf{y}}]) \implies I([\mathbf{x}, Z_{\mathbf{x}}], [\mathbf{y}, Z_{\mathbf{y}}]) = I(\hat{\mathbf{x}}, \hat{\mathbf{y}}) \implies I(\hat{\mathbf{x}}, \hat{\mathbf{y}}) = I(\mathbf{x}, \mathbf{y})$.

In our experiments, we keep the same support dimension for both random variables, increasing it using the following procedure:

1. We sample two binomial random variables $Z_{\mathbf{x}}, Z_{\mathbf{y}}$ with parameters $(n, p)$. We vary $n$ for increasing the complexity of the experiment and we keep $p$ fixed to $0.5$.

2. We concatenate $Z_{\mathbf{x}}$ and $Z_{\mathbf{y}}$ respectively to $\mathbf{x}$ and $\mathbf{y}$, to form higher support versions $\hat{\mathbf{x}}, \hat{\mathbf{y}}$.

3. We map the noisy $\hat{\mathbf{x}}, \hat{\mathbf{y}}$ versions to univariate random variables by applying Cantor's mapping.

#### C.1.2 MODEL AND TRAINING SETUP

In this section, we provide additional experimental details. For all the methods present in the benchmark, we use a Multi Layer Perceptron (MLP) with skip connections and around 130k parameters, based on the architecture used in Franzese et al. (2024), reworking the initial layer to include absolute positional embeddings. For training our discrete score model, we match the methodology used by Lou et al. (2024), using the absorb configuration. At inference time, we always take the last valid validation step estimate of each method to avoid not-a-number values in our tables.

#### C.1.3 STUDY ON TRAINING CONVERGENCE

Since all models were trained in the same conditions, we study the convergence to the correct MI for the benchmarked methods. As we can see from Figure 6, INFO-SEDD is the method which converges faster. Other methods, even when estimating the right MI, take more epochs to settle on the correct value. We also benchmark runtime and maximum memory consumption for the tasks presented in Table 1. We run our experiments on a NVIDIA A100-SXM4-80GB GPU and we observed that INFO-SEDD outperforms all competitors on memory consumption (Table 3) and all variational competitors on runtime (Table 4).

#### C.1.4 LOW DIMENSIONAL BENCHMARK

We also benchmark all methods on an easier low-dimensional benchmark. We use the same training setup, except that we train only for $10^4$ steps using $10^4$ samples and use the same backbone but

| Estimator | INFO-SEDD | GAN-DIME | HD-DIME | KL-DIME | MINDE | MINE | NWJ | SMILE |
|---|---|---|---|---|---|---|---|---|
| MI=10, D=10 | **346.37** | 645.81 | 645.81 | 645.81 | 440.51 | 645.81 | 645.81 | 645.81 |
| MI=20, D=20 | **687.22** | 1301.98 | 1301.98 | 1301.98 | 876.44 | 1301.98 | 1301.98 | 1301.98 |
| MI=30, D=30 | **1005.66** | 1913.15 | 1913.15 | 1913.15 | 1286.35 | 1913.15 | 1913.15 | 1913.15 |
| MI=40, D=40 | **1345.94** | 2564.33 | 2564.33 | 2564.33 | 1726.25 | 2564.32 | 2564.32 | 2564.32 |
| MI=50, D=50 | **1666.30** | 3180.50 | 3180.50 | 3180.50 | 2138.41 | 3180.49 | 3180.49 | 3180.49 |

Table 3: Average maximum memory usage over one epoch for different estimators over different MI estimation tasks. Results are in MB. Lowest memory usages are written in bold.

| Estimator | INFO-SEDD | GAN-DIME | HD-DIME | KL-DIME | MINDE | MINE | NWJ | SMILE |
|---|---|---|---|---|---|---|---|---|
| MI=10, D=10 | 2.40 | 2.89 | 2.90 | 2.86 | **2.16** | 2.89 | 2.96 | 2.94 |
| MI=20, D=20 | 3.34 | 3.99 | 4.02 | 3.98 | **2.62** | 4.03 | 4.03 | 4.00 |
| MI=30, D=30 | 4.51 | 5.56 | 5.54 | 5.57 | **3.37** | 5.57 | 5.57 | 5.61 |
| MI=40, D=40 | 5.96 | 7.34 | 7.33 | 7.36 | **4.00** | 7.39 | 7.42 | 7.37 |
| MI=50, D=50 | 7.06 | 8.71 | 8.69 | 8.71 | **4.68** | 8.76 | 8.73 | 8.72 |

Table 4: Average runtime for one epoch for different estimators over different MI estimation tasks. Results are in seconds. Best runtimes are written in bold.

with 8k parameters. As we can see from Table 5, no clear winner emerges, as all estimators besides MINDE perform a good job in predicting the right MI.

| Estimator | INFO-SEDD | GAN-DIME | HD-DIME | KL-DIME | MINDE | MINE | NWJ | SMILE |
|---|---|---|---|---|---|---|---|---|
| MI=0, D=10 | 0.116 | 0.026 | 0.019 | 0.045 | 2.899 | **0.018** | 0.022 | 0.031 |
| MI=1, D=10 | 0.918 | 1.088 | 0.868 | 0.975 | 2.495 | **0.999** | 1.015 | 1.019 |
| MI=2, D=10 | 1.764 | 2.073 | 1.801 | 1.948 | 2.389 | **2.009** | 1.995 | 2.361 |
| MI=3, D=10 | 2.657 | 2.820 | 3.098 | 3.057 | 3.047 | **2.960** | 2.935 | 3.555 |
| MI=4, D=10 | 4.121 | **4.013** | 3.517 | 4.094 | 3.600 | 4.049 | 3.908 | 4.697 |
| MI=5, D=10 | **4.953** | 4.905 | 4.770 | 4.652 | 4.432 | 4.801 | 4.692 | 2.593 |

Table 5: Results for the low dimensional synthetic benchmark for all estimators. Best estimator in bold.

### C.1.5 SAMPLE COMPLEXITY BENCHMARK

The sample complexity of INFO-SEDD strictly depends on the sample complexity of the score network. Theoretical guarantees on score networks and discrete diffusion models require a careful and involved analysis, which we defer to future work. Nevertheless, we run an ablation study on sample complexity, from an empirical standpoint.

We consider a pair of random vectors $X, Y$, both of length 20. Each component of $X, Y$, in turn, has a support of 4. We set the ground truth MI to $10, 20, 30, 40$ and 50 and use an MLP score network with skip connections and 160k parameters, and train it for 100k steps. We iterate over the number of available training samples $N$ as reported in the following table.

Table 6: Empirical evaluation of sample complexity for INFO-SEDD.

| N | $10^2$ | $10^3$ | $10^4$ | $10^5$ |
|---|---|---|---|---|
| MI=10, D=10 | 7.472 | 11.139 | 10.100 | 10.115 |
| MI=20, D=20 | 7.772 | 22.533 | 20.728 | 19.922 |
| MI=30, D=30 | 26.126 | 29.458 | 36.093 | 29.809 |
| MI=40, D=40 | 28.517 | 38.512 | 35.450 | 38.758 |
| MI=50, D=50 | 27.782 | 50.300 | 43.750 | 48.822 |

INFO-SEDD is sufficiently accurate also for a rather low number of samples. However, as most neural estimators, it relies on deep architectures that require a sufficient number of samples for training. Practitioners are required to find the sweet spot between score network architecture complexity

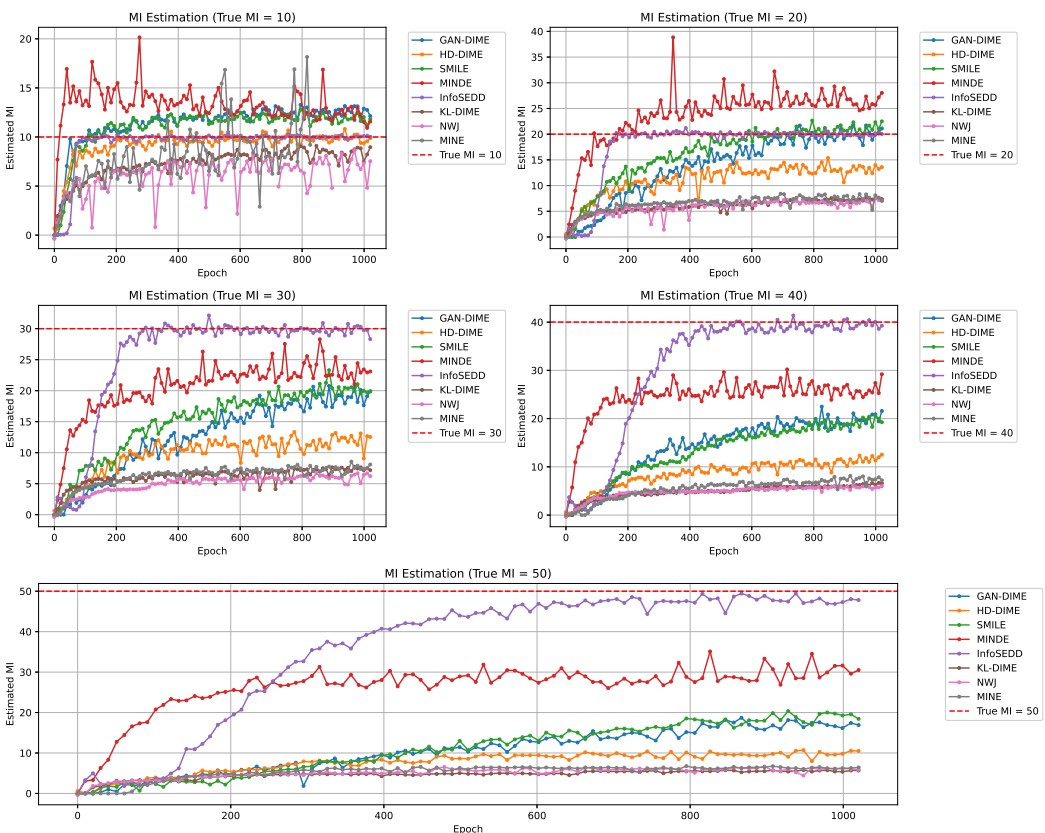

Figure 6: Empirical convergence of the benchmarked MI estimators.

(which caters better MI estimates for complex data) and the need for a large number of training samples.

### C.1.6 ABLATION STUDY ON SUPPORT DIMENSION

In the main paper we performed synthetic experiments by fixing $D$ to 10, MI to 0.5 and increasing $|\chi|$. In this section, we perform the contrary and we increase $\chi$ by using Cantor's mapping and uncorrelated binomial random variables as noise in the data. We increase $\chi|$ as in Table 7, reporting the new support size $\chi$ after the injectio of the noisy binomial random variables. Table 7 shows how INFO-SEDD is the only estimator robust to increasingly high support size.

Table 7: Results of the ablation study on $|\chi|$, with $D$ fixed to 10 and MI=0.5. Best estimates in bold.

| Estimator $\chi$ | INFO-SEDD | GAN-DIME | HD-DIME | KL-DIME | MINDE | MINE | NWJ | SMILE |
|---|---|---|---|---|---|---|---|---|
| 12 | **0.478** | 1.457 | 1.211 | 1.527 | 0.254 | 1.096 | 1.181 | 1.597 |
| 36 | **0.486** | 1.610 | 1.323 | 1.507 | 0.008 | 1.093 | 1.128 | 1.600 |
| 132 | **0.498** | 1.490 | 1.188 | 1.511 | 0.005 | 1.170 | 1.245 | 1.818 |

### C.1.7 EXTREME DIMENSIONALITY EXPERIMENTS

In the main text, we evaluated INFO-SEDD on synthetic benchmarks with increasing dimensionality $D$ and mutual information (MI). Here, we further assess scalability in very high-dimensional regimes and additionally compare against GAN-DIME, HD-DIME, and SMILE for the case where the ground-truth MI is 0.5.

**Case $D = 256$ and MI=0.5** We trained INFO-SEDD using the same architecture as in section 4.1 with an extended budget of $10^6$ optimization steps. Table 8 reports the MI estimates for all methods.

Table 8: Extreme dimensionality experiment with $D = 256$ and ground-truth MI $= 0.5$.

| Method | Estimated MI |
|---|---|
| INFO-SEDD | 0.43 |
| HD-DIME | 0.43 |
| SMILE | 0.67 |
| GAN-DIME | 0.51 |

While competing methods remain relatively stable in this regime, INFO-SEDD provides a competitive and well-calibrated estimate.

**Case $D = 1024$ and MI=0.5** For $D = 1024$, we adopted a DiMamba backbone with 2M parameters, trained for $10^6$ steps with early stopping on a dataset of $10^6$ samples. Results are reported in Table 9.

Table 9: Extreme dimensionality experiment with $D = 1024$ and ground-truth MI $= 0.5$.

| Method | Estimated MI |
|---|---|
| INFO-SEDD | 0.53 |
| HD-DIME | 0.00 |
| SMILE | 0.00 |
| GAN-DIME | 0.00 |

In this extreme regime, all competing methods collapse to zero, whereas INFO-SEDD remains accurate and close to the ground-truth value. These results provide further evidence of the robustness and scalability of our approach in very high-dimensional settings.

**Discussion** These results indicate that INFO-SEDD can be effectively scaled to very high-dimensional discrete distributions when trained with sufficient data and capacity. Together with the experiments in the main paper, they highlight both the robustness of our approach and its ability to leverage larger training budgets when available.

## C.2 TEXT SUMMARIZATION EXPERIMENTS

### C.2.1 DATASET AND GROUND TRUTH CONSTRUCTION

We consider random variable $X$, which correspond to source texts as in Fabbri et al. (2021), and summaries $Y$ obtained with the BART model (M22 of the SummEval paper) when fed with the corresponding source text. Then, for a given $0 < \rho < 1$, we construct $Y^\rho$ which is equal to the original $Y$ with probability $\rho$ and equal to an independent sample from the same distribution of $Y$ with probability $1 - \rho$. Although we cannot establish the ground truth MI $I(X, Y^\rho)$ exactly, we can bound its value. Consider the latent variable $Z$, Bernoulli distributed with parameter $\rho$, which determines whether $Y^\rho$ is equal to the original $Y$ or and independent sample from the same distribution. Then

$$
\begin{aligned}
I(X, Y^\rho) &= I(X; Y^\rho | Z) - I(X; Z | Y^\rho) + I(X, Z) \\
&= I(X; Y^\rho | Z) - I(X; Z | Y^\rho) \\
&\geq I(X; Y^\rho | Z) - H(Z | Y^\rho) \\
&\geq I(X; Y^\rho | Z) - H(Z) \\
&= \rho I(X; Y^\rho | Z = 1) + (1 - \rho) I(X; Y^\rho | Z = 0) - H(Z) \\
&= \rho I(X; Y^\rho | Z = 1) - H(Z) \\
&= \rho I(X; Y) - H(Z)
\end{aligned}
$$

Furthermore, since $I(X, Y^\rho) = I(X; Y^\rho | Z) - I(X; Z | Y^\rho) \leq I(X; Y^\rho | Z) = \rho I(X; Y)$, we can claim that $I_\rho(X, Y) \in [\rho I(X, Y) - H(Z), \rho I(X, Y)]$. Since $H(Z) = -(\rho \log \rho + (1 - \rho) \log(1 - \rho))$, whenever $I(X; Y)$ is much greater than $H(Z)$, $I(X; Y^\rho)$ satisfies approximately the linear relationship $I(X; Y^\rho | Z) = \rho I(X, Y)$. This assumption is supported by the literature (Takahira et al., 2016).

### C.2.2 MODEL SETUP

For all our methods, we use the MDLM-small model (Sahoo et al., 2024) trained on OpenWebText. While this family of models does not directly parametrize the score, Sahoo et al. (2024) show that it can be easily recovered from the output of the model. For the experiments with MINDE, we use the variance preserving Stochastic Differential Equation (SDE) and the same model. We perform the diffusion step on the token embeddings. We trained all models using the configuration in Table 10.

Table 10: Training settings for one run for the model selection experiments and the consistency tests (one value of $\rho$)

| Parameter | Value |
|---|---|
| Backbone | MDLM small (Sahoo et al., 2024) |
| Optimizer | AdamW |
| AdamW - $\beta_1$ | 0.9 |
| AdamW - $\beta_2$ | 0.999 |
| Weight decay | 0 |
| Learning rate | 3e-4 |
| Batch size | 64 |
| Number of steps | 3600 |
| Dropout rate | 0.0 |
| Hardware | 1×NVIDIA A 100 80 GB |

### C.2.3 NUMERICAL RESULTS

In Table 11, we report the numerical results for MI estimations in the consistency tests, including MINDE. We report all values in nats.

Table 11: Numerical results for Text summarization consistency tests

| $\rho$ | GAN-DIME | HD-DIME | INFO-SEDD-C | INFO-SEDD-J | KL-DIME | MINDE | MINE | NWJ | SMILE |
|---|---|---|---|---|---|---|---|---|---|
| 0.0 | - | 0.09 | 7.52 | 106.69 | -0.01 | 365679296.00 | -0.50 | -0.01 | 0.00 |
| 0.1 | - | 0.20 | 34.02 | 157.90 | 0.01 | 198489552.00 | -0.02 | 0.00 | 0.00 |
| 0.2 | - | -0.01 | 61.25 | 197.18 | 0.48 | 347922112.00 | 0.02 | -0.10 | 0.09 |
| 0.3 | - | 0.13 | 90.17 | 220.47 | 3.03 | 128288973.00 | 0.00 | -0.24 | 0.89 |
| 0.4 | - | 2.41 | 124.48 | 244.53 | 3.21 | 33884712.00 | 0.84 | - | 2.06 |
| 0.5 | - | 10.57 | 170.03 | 255.42 | 4.63 | 19269294.00 | 0.33 | 0.19 | 5.84 |
| 0.6 | - | 16.17 | 230.41 | 276.28 | 5.26 | 270945856.00 | -4.11 | 0.84 | 7.19 |
| 0.7 | 9.63 | 12.94 | 308.30 | 303.20 | 5.17 | 391443968.00 | -3.68 | - | 12.46 |
| 0.8 | 9.28 | 28.98 | 409.26 | 316.01 | 6.38 | 448679872.00 | -3.90 | - | 13.85 |
| 0.9 | 9.33 | 27.12 | 505.13 | 330.76 | 7.95 | 18818858.00 | -3.96 | - | 14.73 |
| 1.0 | 19.44 | 39.42 | 602.77 | 361.23 | 6.82 | 1242223336.00 | -3.96 | - | 20.55 |

### C.2.4 DETAILED RESULTS FOR HUMAN METRICS CORRELATION EXPERIMENTS

We report the MI estimates of each model and compare them with human evaluation metrics. For each metric, we run a GP regression with a Matérn kernel with smoothing parameter $\nu = 1.5$ (Rasmussen and Williams, 2006) and noise level $0.01$ using the MI estimates from both INFO-SEDD-C and INFO-SEDD-J. We use the *scikit-learn* Python package to run this analysis (Pedregosa et al., 2011). In Figures 7 and 8, we plot the GP mean, its 95% confidence interval, and the Pearson correlation. As the figures show, several human metrics saturate around 5.0 (the maximum score possible), while MI is not subject to this ceiling effect.

In particular, for the consistency and fluency metrics, the GP mean follows an increasing trend until the human scores saturate, suggesting that the linear relationship between MI and the human metric only holds up to a threshold, beyond which most models achieve the maximum score.

Unlike human metrics, however, INFO-SEDD does not rely on annotations: it only requires a dataset of summaries paired with source texts, making it a cost-effective and scalable alternative for model selection in text summarization.

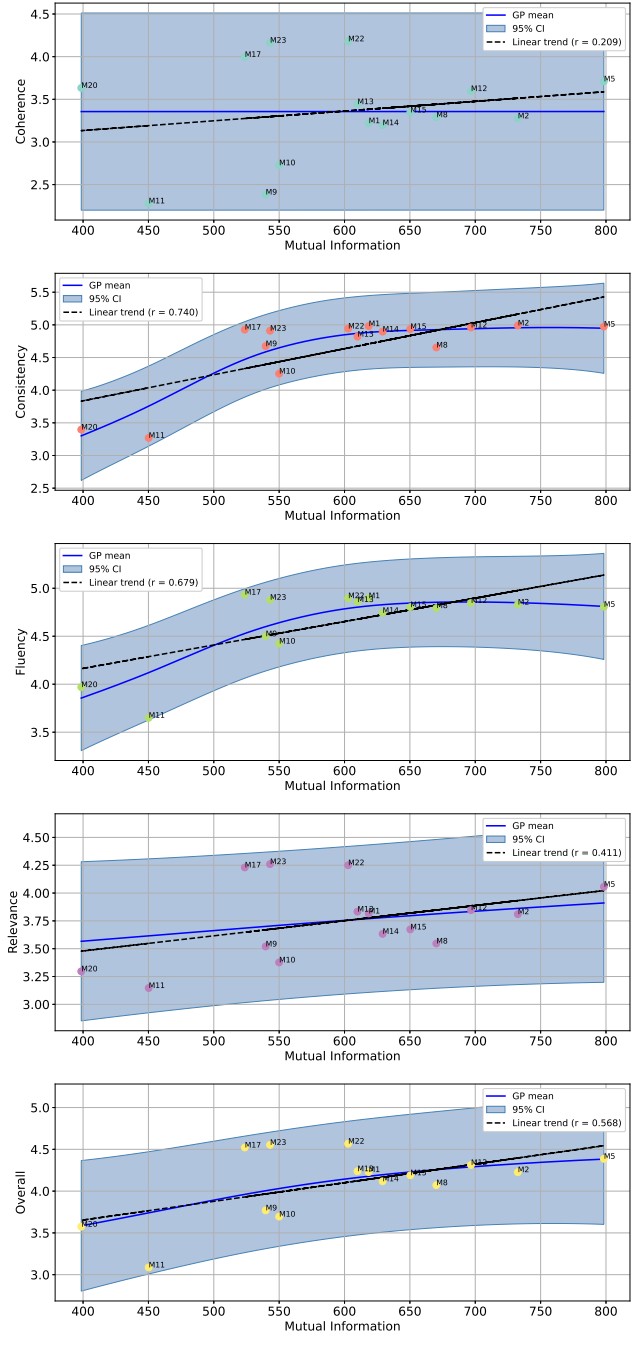

Figure 7: Comparison of human metrics against mutual information estimated by INFO-SEDD-C

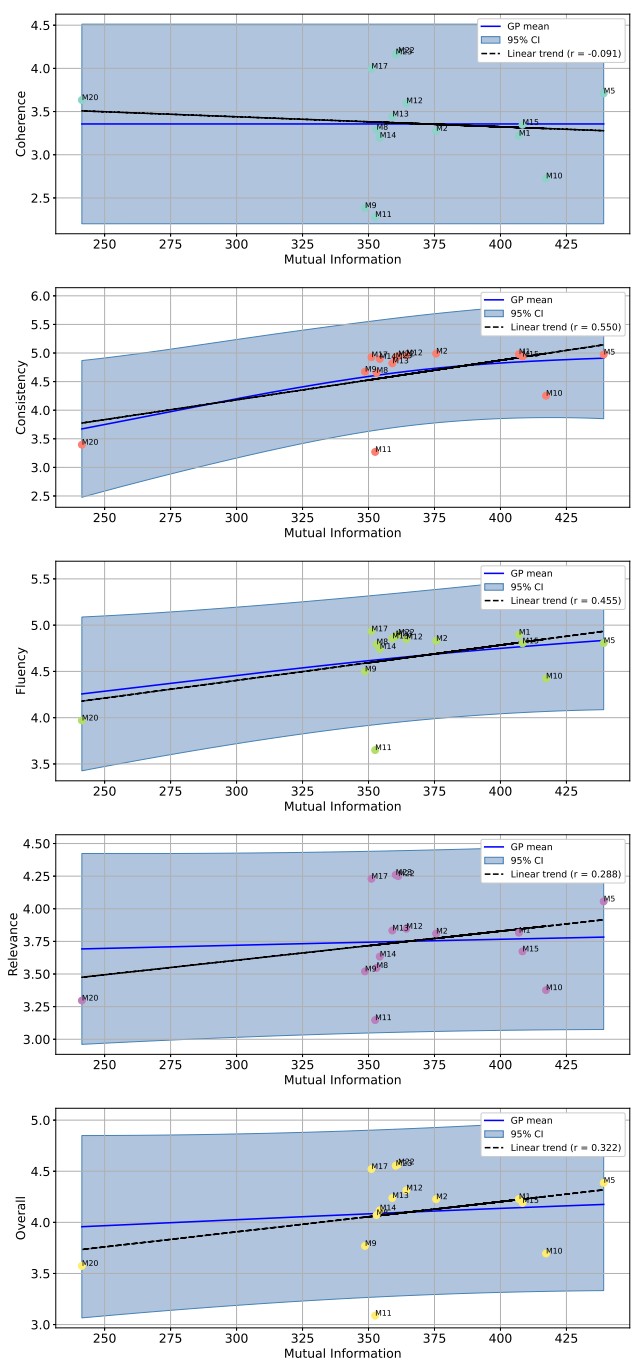

Figure 8: Comparison of human metrics against mutual information estimated by INFO-SEDD-J

### C.3 COMPARISON OF MI ESTIMATES WITH INFO-SEDD AND OTHER TEXT SUMMARIZATION METRICS

We measured several other metrics using the SummEval library (Fabbri et al., 2021) and one reference summary for each sample provided in the SummEval collection. In particular, we benchmark the BERT SCORE metric (Zhang* et al., 2020), the BLEU metric (Papineni et al., 2002) and the CIDER metric (Vedantam et al., 2015). As we can see from Figure 10 and Figure 9, we find that INFO-SEDD-J in general correlates better with other summarization metrics with respect to INFO-SEDD-C. Interestingly, INFO-SEDD-C correlates better with human metrics, proving to be a useful

proxy for human metrics without the need of human annotations. Notably, INFO-SEDD-C is also the best at correlating with consistency among all other text summarization metrics.

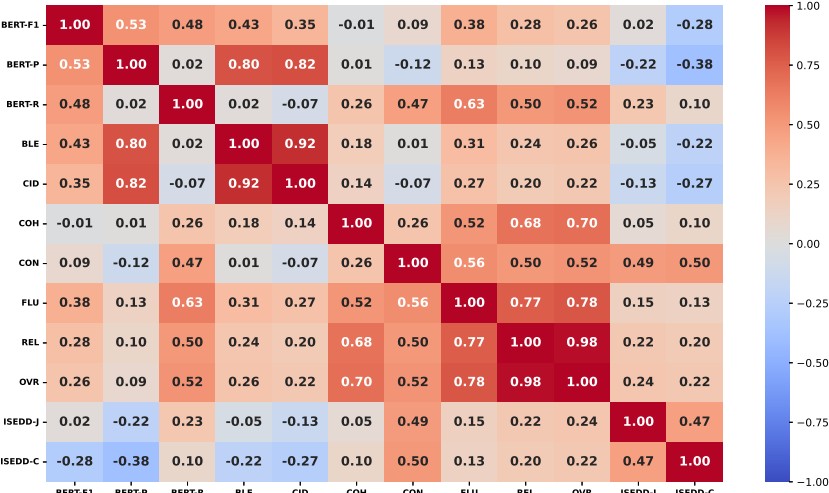

Figure 9: Kendall Tau correlation heatmap of MI estimates INFO-SEDD, other text summarization metrics and human metrics. CID=Cider, BLE=bleu.

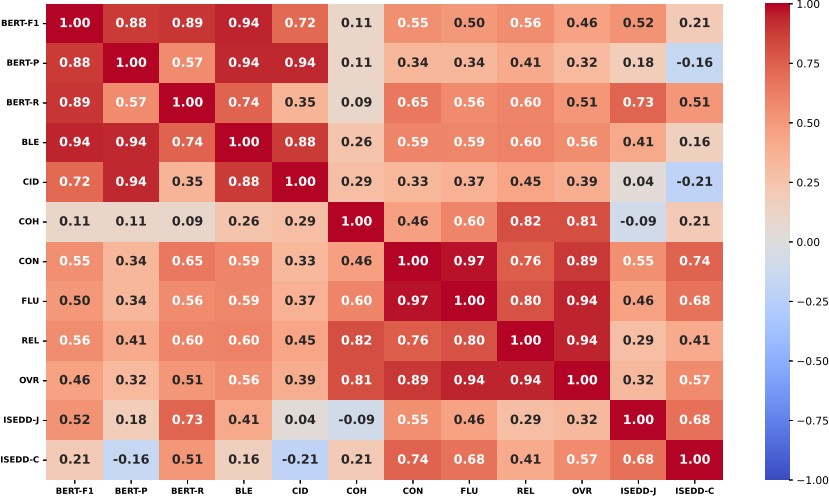

Figure 10: Pearson correlation heatmap of MI estimates INFO-SEDD, other text summarization metrics and human metrics. CID=Cider, BLE=bleu.

## C.4 GENOMICS EXPERIMENTS

### C.4.1 DATASET AND GROUND TRUTH CONSTRUCTION

To construct our dataset for the consistency test, we consider the shuffling hyperparameter $\rho$. We consider $X$ to be the random variable representing the DNA sequences and, for a given $0 < \rho < 1$, we construct $Y^\rho$ which is equal to the original $Y$, the random variable representing the human or worm label, with probability $1 - \rho$ and equal to an independent sample from the same distribution of $Y$ with probability $\rho$. We can estimate $I(X, Y^\rho)$ by considering the entropy terms which correspond to the decomposition $H(Y^\rho) - H(Y^\rho|X)$. On the one hand, estimation of $H(Y^\rho) = H(Y)$ from the dataset is relatively simple and stable given the low dimensionality of the label. Concerning $H(Y^\rho|X)$, we first write the conditional distribution $p_{Y^\rho|X}(y^\rho|x)$ as:

$$p_{Y^\rho|X}(y^\rho|x) = \sum_y p_{Y^\rho,Y|X}(y^\rho, y|x)$$

$$= \sum_y p_{Y^\rho|Y,X}(y^\rho|y, x)p_{Y|X}(y|x)$$

$$= \sum_y (1 - \rho)\delta(y, y^\rho)p_{Y|X}(y|x) + \rho p_Y(y^\rho)p_{Y|X}(y|x)$$

$$= (1 - \rho)p_{Y|X}(y^\rho|x) + \rho p_Y(y^\rho)$$

In the considered dataset, classes are balanced and consequently $p_Y(y^\rho) = 0.5$. To approximate $p_{Y|X}(y^\rho|x)$, we first define $l(x) = \arg\max_y p_{Y|X}(y|x)$, then we assume the following:

$$p_{Y|X}(y|x) = \begin{cases} 1 - \epsilon & \text{for } y = l(x) \\ \epsilon & \text{otherwise} \end{cases}$$

with $\epsilon \ll 1$, which roughly speaking corresponds to assuming that for all values of $x$ the conditional probability is highly skewed either towards zero or one. This assumption proves to be reasonable in scenarios where the classification accuracy is close to one, as the case considered here (Schiff et al., 2024). Then

$$p_{Y^\rho|X}(y^\rho|x) = \begin{cases} (1 - \rho)(1 - \epsilon) + \rho p_Y(y^\rho) & \text{for } y = l(x) \\ (1 - \rho)\epsilon + \rho p_Y(y^\rho) & \text{otherwise} \end{cases}$$

Going back to $H(Y^\rho|X)$, we can finally write:

$$H(Y^\rho|X) = -\sum_x p_X(x) \sum_y p_{Y^\rho|X}(y^\rho|x) \log p_{Y^\rho|X}(y^\rho|x)$$

$$= -\sum_x p_X(x)(p_{Y^\rho|X}(l(x)|x) \log p_{Y^\rho|X}(l(x)|x) +$$

$$+ p_{Y^\rho|X}(1 - l(x)|x) \log p_{Y^\rho|X}(1 - l(x)|x))$$

$$= \sum_x p_X(x)((1 - \rho)(1 - \epsilon) + \rho p_Y(y^\rho)) \log((1 - \rho)(1 - \epsilon) + \rho p_Y(y^\rho))$$

$$+ \sum_x p_X(x)((1 - \rho)\epsilon + \rho p_Y(y^\rho)) \log((1 - \rho)\epsilon + \rho p_Y(y^\rho))$$

$$= \sum_x p_X(x)H_b((1 - \rho)(1 - \epsilon) + \rho 0.5) = H_b((1 - \rho)(1 - \epsilon) + \rho 0.5)$$

To select $\epsilon$, we choose the accuracy of the best classifier from (Schiff et al., 2024) trained for the HUMAN VS WORM classification task.

### C.4.2 MODEL SETUP

We employed a pretrained Caduceus model (Schiff et al., 2024) as a backbone for all estimators. We modified the head of the model when needed. For MINE-like estimators we added an attention head

and added a fully connected layer for aggregating the output. For MINDE, we added a transformer block with time conditioning. For INFO-SEDD, we did not make architectural variations and we employ the optimized training strategy highlighted in Appendix C.1. We trained all models using the configuration in Table 12. For the motif selection experiments, we employ INFO-SEDD-J because our method requires masking elements of the DNA sequence, which is something that INFO-SEDD-C does not do during training.

Table 12: Training settings for one run for the promoter experiments or the consistency tests (one value of $\rho$)

| Parameter | Value |
|---|---|
| Backbone | Caduceus-1K (Schiff et al., 2024) |
| Optimizer | AdamW |
| AdamW - $\beta_1$ | 0.9 |
| AdamW - $\beta_2$ | 0.999 |
| Weight decay | 0.1 |
| Learning rate | 0.001 |
| Batch size | 128 |
| Number of steps | 20000 |
| Dropout rate | 0.1 |
| Hardware | 1×NVIDIA A 100 80 GB |

### C.4.3 NUMERICAL RESULTS FOR THE CONSISTENCY TESTS

In Table 13, we report the numerical results for MI estimations in the consistency tests, including MINDE. We report all values in nats.

Table 13: Numerical results for DNA consistency tests

| $\rho$ | GAN-DIME | HD-DIME | INFO-SEDD-C | INFO-SEDD-J | KL-DIME | MINE | NWJ | SMILE |
|---|---|---|---|---|---|---|---|---|
| 0.0 | 0.01 | 0.00 | 0.00 | 0.05 | 0.00 | 0.00 | -0.02 | 0.00 |
| 0.1 | 0.00 | 0.01 | 0.00 | 0.07 | 0.00 | 0.00 | -0.01 | 0.00 |
| 0.2 | 0.02 | -0.01 | 0.00 | 0.06 | 0.00 | 0.00 | -0.02 | 0.02 |
| 0.3 | 0.04 | 0.01 | 0.03 | 0.08 | 0.00 | 0.00 | -0.01 | 0.02 |
| 0.4 | 0.00 | 0.07 | 0.07 | 0.06 | -0.01 | 0.00 | 0.00 | 0.07 |
| 0.5 | 0.04 | 0.14 | 0.12 | 0.09 | 0.12 | 0.13 | 0.13 | 0.12 |
| 0.6 | 0.18 | 0.16 | 0.18 | 0.09 | 0.17 | 0.19 | -0.02 | 0.19 |
| 0.7 | 0.27 | 0.23 | 0.24 | 0.14 | 0.30 | 0.29 | 0.00 | 0.11 |
| 0.8 | 0.34 | 0.33 | 0.32 | 0.18 | 0.00 | 0.38 | -0.01 | 0.29 |
| 0.9 | 0.15 | 0.43 | 0.45 | 0.24 | 0.51 | 0.54 | -0.02 | 0.40 |
| 1.0 | 0.69 | -0.01 | 0.61 | 0.37 | 0.66 | 0.71 | 0.70 | 0.50 |

# D ADDITIONAL RESULTS

## D.1 ISING MODEL ENTROPY ESTIMATION

The Ising model is a system consisting of particles arranged in a lattice. In our experiments, we consider a $L \times L$ square. A particle $i$ of the lattice is associated with a discrete value $\sigma_i \in \{-1, +1\}$ called spin and each pair of particles $ij$ is characterized by an interaction strength $J_{ij}$. With no external fields, these quantities determine the energy $E(\sigma)$ of the configuration $\sigma$:

$$E(\sigma) = \sum_{i,j} J_{ij} \sigma_i \sigma_j$$

In turns, the energy of a configuration determines its likelihood. In particular, the configurations of the Ising model follow a probability distribution $\vec{p}_0$ parametrised by the temperature $T$, the Boltzmann constant $k_b$ and the interaction strengths:

$$\vec{p}_0(\sigma) = \frac{e^{-\beta E(\sigma)}}{Z(T)}$$

Where $Z(T) = \sum_i e^{-\beta E(\sigma_i)}$ and $\beta = (k_b T)^{-1}$. In order to generate our dataset from $\vec{p}_0$, we follow the Metropolis algorithm (Bhanot, 1988):

---

**Algorithm 4** Metropolis Algorithm for 2D Ising Spin Glass

---

1: **Input:** Lattice size $N$, interaction strengths $J_{ij}$, temperature $T$, number of iterations iter_max
2: **Initialize:** Spin lattice $\sigma$ with $\sigma_{i,j} \in \{-1, +1\}$ randomly assigned
3: **for** iteration = 1 to iter_max **do**
4:     Randomly select a lattice site $i$
5:     Compute the change in energy $\Delta E$ if $\sigma_{i,j}$ is flipped:

$$\Delta E = 2\,\sigma_i \sum_j J_{i,j}\,\sigma_j$$

6:     Generate a random number $r$ uniformly distributed in $[0, 1]$
7:     **if** $r < \exp(-\beta \Delta E)$ **then**
8:         Flip the spin: $\sigma_i \leftarrow -\sigma_i$
9:     **end if**
10: **end for**
11: **Output:** Final spin configuration $\sigma$

---

We compute the entropy of $\vec{p}_0$ analytically, starting from the free energy $F$ per site of the lattice:

$$F(T) = -k_b T \log \lambda_T$$

Where $\lambda_T$ is the partition function, which depends on the interaction horizontal and vertical interaction strength. For simplicity, we consider the same interaction strength $J = 1$ for all neighboring particles, while we set it to zero for non neighboring particles. Under these assumptions, we can calculate $\log \lambda$ with a double integral (Onsager, 1944):

$$\log \lambda_T = \log 2 + \frac{1}{2\pi^2} \int_0^\pi \int_0^\pi \log(\cosh(2\beta J)\cosh(2\beta J) - \sinh(2\beta J)\cos(\theta_1) - \sinh(2\beta J)\cos(\theta_2)) d\theta_1 d\theta_2$$

For simplicity, we also set $k_b = 1$. From $F$, we can calculate the entropy $H$ using the thermodynamic relation $H = -\frac{\partial F}{\partial T}$. We compute the integral numerically using the *SciPy* Python package (Virtanen et al., 2020) and we approximate $H$ as $H \approx \frac{F(T+\Delta T) - F(T-\Delta T)}{2\Delta T}$, with $\Delta T = 10^{-4}$.

For what concerns the INFO-SEDD architecture, we keep a single model configuration for all temperatures, both for the model, which contains around $90k$ parameters, and for the diffusion. To get the

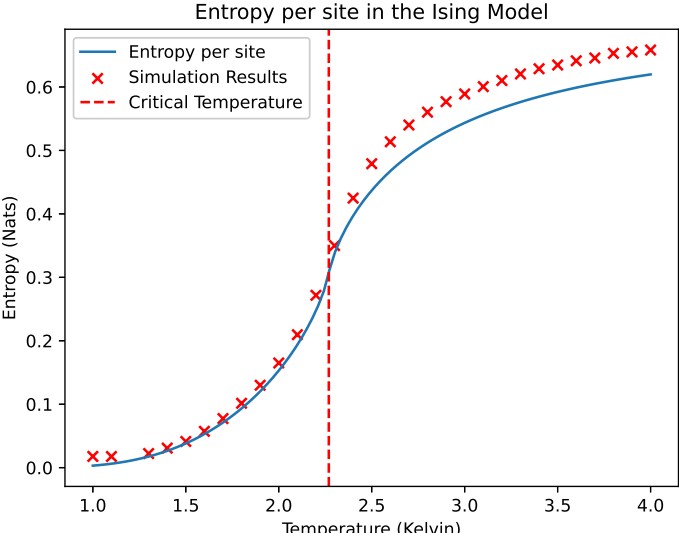

Figure 11: Ising model entropy estimates

entropy per site, we divide the estimates of the model by the number of particles in the configurations (400).

In this section, we numerically validate the performance of INFO-SEDD by estimating entropy according to Algorithm 3.

**Spin glasses experiments.**   Entropy computation in Ising models enables insights on the thermodynamics properties of the system (Cincio et al., 2007), which can be used for scientific discovery in the domain where the Ising model is applied (Macy et al., 2024; Schneidman et al., 2006; Sherrington and Kirkpatrick, 1975).

**Experimental setup**   We consider a simplified Ising model applied to spin glasses (Sherrington and Kirkpatrick, 1975). We do not include an external field, we set a unitary interaction strength for all the sites interactions, unitary Boltzmann's constant and a $20 \times 20$ square lattice. The entropy per site of this configuration can be computed in closed form Onsager (1944). We test our model by estimating the entropy per particle at linearly spaced temperatures from $1.0K$ to $4.0K$. We generate our dataset using the Metropolis-Hastings algorithm, with $10000$ samples for each temperature (see Algorithm 4). We post-process the output of INFO-SEDD by dividing the entropy estimate by $400$ to report the entropy per site.

**Results and Analysis**   Variational estimators cannot estimate large KL divergences reliably with limited samples sizes (McAllester and Stratos, 2020; Song and Ermon, 2019a). In this scenario, instead, INFO-SEDD accurately estimates large KL divergences (Figure 11), performing particularly well at low temperatures where we need to estimate large KL divergences.

## D.2   COMPARISON WITH PLUG-IN ESTIMATORS

In this section, we discuss the limitations of plug-in estimators for discrete random variables. First, we analyze the approach by Goebel et al. (2005), where the authors use a Taylor expansion to estimate MI. However, by construction, this estimator assumes a low MI regime. This assumption makes the method inherently unsuitable for high-MI estimation tasks, where higher-order terms in the expansion become non-negligible and induce severe bias.

We further analyze the approach of Gao et al. (2017), who propose a variation of the KSG estimator to handle mixtures of continuous and discrete data. We benchmark this estimator on the

high-dimensional synthetic benchmark, as in Table 1, and on the extreme high-dimensional setting introduced in Appendix C.1.7. Results are reported in Tables 14 and 15.

Table 14: Plug-in estimator (Gao et al. (2017)) on high-MI synthetic benchmarks with $D = \mathrm{MI}$.

| Setting | Estimated MI |
|---|---|
| MI=10, D=10 | 11.56 |
| MI=20, D=20 | 9.33 |
| MI=30, D=30 | 10.55 |
| MI=40, D=40 | 2.87 |
| MI=50, D=50 | 4.18 |

We observe that while the estimator remains reasonably accurate for moderate MI (e.g., MI=10), its performance quickly deteriorates as dimensionality and dependence strength increase.

We additionally evaluate the estimator in the extreme dimensionality setting with low ground-truth MI (0.5), as introduced in Appendix C.1.7.

Table 15: Plug-in estimator (Gao et al. (2017)) in extreme dimensionality regimes with ground-truth $\mathrm{MI} = 0.5$.

| MI | D | Estimated MI |
|---|---|---|
| 0.5 | 256 | 14.17 |
| 0.5 | 1024 | 14.30 |

In this regime, the estimator overestimates the mutual information. This behavior highlights the instability of plug-in approaches in high-dimensional discrete settings.

Overall, these experiments confirm that plug-in estimators based on Taylor approximations or nearest-neighbor corrections are not robust to either high-MI or high-dimensional regimes. In contrast, INFO-SEDD remains stable and accurate across both challenging scenarios.

# E    THEORETICAL PROPERTIES OF INFO-SEDD

We highlight the main assumptions and setting in Appendix E.1, then in Appendix E.2 we proceed to construct an error bound on the KL estimate of INFO-SEDD under the established setting.

## E.1    ASSUMPTIONS AND SETTING

We highlight the assumptions on the probability distributions and score networks. We first define the class of functions $\mathcal{F}$ which restricts the class of score functions and score networks. In this work, we consider CTMC with only sparse transition rate matrices $\overrightarrow{Q}$, which constrain the CTMC to modify only one subcomponent at a time. This means that, when considering the KL estimate as in Equation (4), only ratios of probability distributions $\frac{\overrightarrow{p}_t(a)}{\overrightarrow{p}_t(b)}$ with $a$ and $b$ within Hamming distance 1 are considered. The assumptions that we are going to make in this section reflect this characteristic. We define the set of pairs of samples at Hamming distance 1 as follows:

$$\mathcal{S} := \{(x', x'') \in \chi^D \times \chi^D \mid d_{\text{hamming}}(x', x'') = 1\}.$$

Where $\chi^D$ is the support of $D$-dimensional random vectors We impose two finite constants

$$0 < C_1 < C_2 < +\infty,$$

and define

$$\mathcal{F} := \{f : \mathcal{S} \times [0, T] \to \mathbb{R} \mid C_1 \leq ||f||_\infty \leq C_2\},$$

where $||f||_\infty$ denotes the supremum norm over $\mathcal{S} \times [0, T]$.

Throughout this appendix, we use a family of neural networks $T_\Theta \subset \mathcal{F}$ that are standard multilayer feedforward networks with bounded, nonconstant activation functions (Hornik et al., 1989). We also fix a perturbation kernel $\overrightarrow{p}_{t|0}$ and denote the score function for arbitrary probability distribution $p$ as

$$s^p(x_t)_{\hat{x}} = \frac{\overrightarrow{p}_t(\hat{x})}{\overrightarrow{p}_t(x_t)},$$

where $\overrightarrow{p}_t(x_t) = \sum_{x_0} \overrightarrow{p}_{t|0}(x_t|x_0)p(x_0)$.

We now state the assumptions on score networks and on score functions:

- **A.1**: For all $s_\theta^p \in T_\Theta$, $C_1 \leq ||s_\theta^p||_\infty \leq C_2$.
- **A.2**: For all $p$, the score function satisfies $C_1 \leq ||s^p||_\infty \leq C_2$.

The lower bound $C_1 > 0$ avoids degeneracies, and ensures well-defined log terms in the loss. We define the class of distribution $\mathcal{P}$ for which **A.2** is true as

$$\mathcal{P} := \{p \mid C_1 \leq ||s^p||_\infty \leq C_2\}$$

**A.1** and **A.2** are similar to Assumption 4.4 of Ren et al. (2025), who assume that both the score function and the score networks are bounded. Similarly, Chen and Ying (2025) set bound for the score network (Assumption 4.2) and for the true score function (Assumption 4.3).

## E.2    ERROR ANALYSIS OF INFO-SEDD

Let $p, q \in \mathcal{P}$ be two probability distributions satisfying the assumptions highlighted in Appendix E.1. Let $\theta, \phi \in \Theta$ be two arbitrary sets of parameters for the score networks $s_\theta^p, s_\phi^q$.

$$\epsilon_p = ||s_\theta^p - s^p||_\infty, \qquad \epsilon_q = ||s_\phi^q - s^q||_\infty.$$

Then, let us define the functional $\mathcal{E}$, parameterised with arbitrary functions $f, g \in \mathcal{F}$ and perturbation kernel $\overrightarrow{p}_{t|0}$ as

$$\mathcal{E}(f, g; x_0) = \int_0^T \mathbb{E}_{x_t \sim \overrightarrow{p}_{t|0}(\cdot|x_0)} \sum_{\hat{x} \neq x} \overrightarrow{Q}(x_t, \hat{x})(K(f(x_t)_{\hat{x}}) + f(x_t)_{\hat{x}} - g(x_t)_{\hat{x}} \log g(x_t)_{\hat{x}}) dt$$

Then, the INFO-SEDD estimator for the KL is

$$\mathbb{E}_{x \sim p} \mathcal{E}(s_\theta^p, s_\phi^q; x)$$

The true KL divergence, instead is exactly

$$\mathrm{KL}\left[p \parallel q\right] = \mathrm{KL}\left[\overrightarrow{p}_T \parallel \overrightarrow{q}_T\right] + \mathbb{E}_{x \sim p} \mathcal{E}\left(s^p, s^q; x\right)$$

Then, we chracterise the error of INFO-SEDD as

$$\left|\mathbb{E}_{x \sim p} \mathcal{E}(s_\theta^p, s_\phi^q; x) - \mathrm{KL}\left[\overrightarrow{p}_T \parallel \overrightarrow{q}_T\right] - \mathbb{E}_{x \sim p} \mathcal{E}\left(s^p, s^q; x\right)\right| \leq \left|\mathbb{E}_{x \sim p} \mathcal{E}(s_\theta^p, s_\phi^q; x) - \mathbb{E}_{x \sim p} \mathcal{E}\left(s^p, s^q; x\right)\right| + \mathrm{KL}\left[\overrightarrow{p}_T \parallel \overrightarrow{q}_T\right] \tag{11}$$

We denote $\emptyset^D$ the element in $\chi^D$ such that $\emptyset_i^D = \emptyset \, \forall i$ and we now focus first on the second term of the right hand side of Equation (11)

$$\mathrm{KL}\left[\overrightarrow{p}_T \parallel \overrightarrow{q}_T\right] = \sum_x \overrightarrow{p}_T(x) \log \frac{\overrightarrow{p}_T(x)}{\overrightarrow{q}_T(x)}$$

Firstly, note that for any $t \in (0, T]$, $\overrightarrow{p}_t(\emptyset^D) = \overrightarrow{q}_t(\emptyset^D)$, and as a consequence $\log \frac{\overrightarrow{p}_t(\emptyset^D)}{\overrightarrow{q}_t(\emptyset^D)} = \log(1) = 0$. This means that we can write

$$\begin{aligned}
\mathrm{KL}\left[\overrightarrow{p}_T \parallel \overrightarrow{q}_T\right] &= \sum_x \overrightarrow{p}_T(x) \log \frac{\overrightarrow{p}_T(x)}{\overrightarrow{q}_T(x)} \\
&= \sum_{x \neq \emptyset^D} \overrightarrow{p}_T(x) \log \frac{\overrightarrow{p}_T(x)}{\overrightarrow{q}_T(x)} \\
&\leq \sum_{x \neq \emptyset^D} \overrightarrow{p}_T(x) \log \frac{1}{\overrightarrow{q}_T(x)} \\
&= \sum_{x \neq \emptyset^D} \overrightarrow{p}_T(x) \log \sum_{\hat{x}} \frac{\overrightarrow{q}_T(\hat{x})}{\overrightarrow{q}_T(x)}.
\end{aligned}$$

Note that we cannot bound $\frac{\overrightarrow{q}_T(\hat{x})}{\overrightarrow{q}_T(x)}$ by simply applying **A.2**, since the sum involves terms such that $d_{\mathrm{hamming}}(x, \hat{x}) > 1$. We can however construct an ordered sequence of points $S(x, \hat{x})$ where $x$ is the first element, $\hat{x}$ is the last element and $\forall i, j$ such that $0 \leq i < j < d_{\mathrm{hamming}}(x, \hat{x})$ it holds $d_{\mathrm{hamming}}(S(x, \hat{x})_j, S(x, \hat{x})_i) = j - i$. This allows us to write

$$\begin{aligned}
\frac{\overrightarrow{q}_T(\hat{x})}{\overrightarrow{q}_T(x)} &= \prod_{i=0}^{d_{\mathrm{hamming}}(x, \hat{x})-2} \frac{\overrightarrow{q}_T(S(x, \hat{x})_{i+1})}{\overrightarrow{q}_T(S(x, \hat{x})_i)} \\
&= \prod_{i=0}^{d_{\mathrm{hamming}}(x, \hat{x})-2} s^q(S(x, \hat{x})_i)_{S(x, \hat{x})_{i+1}} \\
&\leq \prod_{i=0}^{d_{\mathrm{hamming}}(x, \hat{x})-2} C_2 \\
&\leq C_2^D
\end{aligned}$$

We can now go back to $\text{KL}\left[\overrightarrow{p}_T \parallel \overrightarrow{q}_T\right]$,

$$
\begin{aligned}
\text{KL}\left[\overrightarrow{p}_T \parallel \overrightarrow{q}_T\right] &\le \sum_{x \ne \emptyset^D} \overrightarrow{p}_T(x) \log \sum_{\hat{x}} \frac{\overrightarrow{q}_T(\hat{x})}{\overrightarrow{q}_T(x)} \\
&\le \sum_{x \ne \emptyset^D} \overrightarrow{p}_T(x) \log \sum_{\hat{x}} C_2^D \\
&= \sum_{x \ne \emptyset^D} \overrightarrow{p}_T(x) \log |\chi|^D C_2^D \\
&= (1 - \overrightarrow{p}_T(\emptyset^D))(D \log |\chi| C_2).
\end{aligned}
$$

We can now focus on the first term of the right hand side of Equation (11)

$$
\left| \mathbb{E}_{x \sim p} \mathcal{E}(s_\theta^p, s_\phi^q; x) - \mathbb{E}_{x \sim p} \mathcal{E}\left(s^p, s^q; x\right) \right|
$$

First, we decompose each term and use triangle inequality

$$
\begin{aligned}
&\left| \mathbb{E}_{x \sim p} \mathcal{E}(s_\theta^p, s_\phi^q; x) - \mathbb{E}_{x \sim p} \mathcal{E}(s^p, s^q; x) \right| \\
&= \left| \mathbb{E}_{x \sim p}\left( \mathcal{E}(s_\theta^p, s_\phi^q; x) - \mathcal{E}(s^p, s^q; x) \right) \right| \\
&= \left| \mathbb{E}_{x \sim p} \int_0^T \mathbb{E}_{x_t \sim \overrightarrow{p}_{t|0}(\cdot|x_0)} \sum_{\hat{x} \ne x} \overrightarrow{Q}(x_t, \hat{x})\left( K\!\left(s_\theta^p(x_t)_{\hat{x}}\right) - K(s^p(x_t)_{\hat{x}}) \right.\right. \\
&\qquad\qquad + s_\phi^q(x_t)_{\hat{x}} - s^q(x_t)_{\hat{x}} - s_\theta^p(x_t)_{\hat{x}} \log\!\left(s_\phi^q(x_t)_{\hat{x}}\right) \\
&\qquad\qquad \left.\left. + s^p(x_t)_{\hat{x}} \log(s^q(x_t)_{\hat{x}}) \right) dt \right| \\
&\le \mathbb{E}_{x \sim p} \int_0^T \mathbb{E}_{x_t \sim \overrightarrow{p}_{t|0}(\cdot|x_0)} \sum_{\hat{x} \ne x} \overrightarrow{Q}(x_t, \hat{x})\left( \left| K\!\left(s_\theta^p(x_t)_{\hat{x}}\right) - K(s^p(x_t)_{\hat{x}}) \right| \right. &&(12) \\
&\qquad\qquad + \left| s_\phi^q(x_t)_{\hat{x}} - s^q(x_t)_{\hat{x}} \right| \\
&\qquad\qquad \left. + \left| s_\theta^p(x_t)_{\hat{x}} \log\!\left(s_\phi^q(x_t)_{\hat{x}}\right) - s^p(x_t)_{\hat{x}} \log(s^q(x_t)_{\hat{x}}) \right| \right) dt
\end{aligned}
$$

We use the inequality $\log(a) \leq a - 1$ to proceed with the first term of Equation (12),

$$\left| K\big(s_\theta^p(x_t)_{\hat{x}}\big) - K\big(s^p(x_t)_{\hat{x}}\big) \right| = \left| s_\theta^p(x_t)_{\hat{x}}\big(\log(s_\theta^p(x_t)_{\hat{x}}) - 1\big) - \big(s^p(x_t)_{\hat{x}}\big)\big(\log\big(s^p(x_t)_{\hat{x}}\big) - 1\big) \right|$$

$$\leq \left| s_\theta^p(x_t)_{\hat{x}} \log(s_\theta^p(x_t)_{\hat{x}}) - \big(s^p(x_t)_{\hat{x}}\big)\log\big(s^p(x_t)_{\hat{x}}\big) \right| + \epsilon_p$$

$$\leq \max(s^p(x_t)_{\hat{x}}, s_\theta^p(x_t)_{\hat{x}}) \left| \log(s_\theta^p(x_t)_{\hat{x}}) - \log\big(s^p(x_t)_{\hat{x}}\big) \right| + \epsilon_p$$

$$\leq C_2 \left| \log\left( \frac{s_\theta^p(x_t)_{\hat{x}}}{s^p(x_t)_{\hat{x}}} \right) \right| + \epsilon_p$$

$$\leq C_2 \left| \left( \frac{\max(s_\theta^p(x_t)_{\hat{x}}, s^p(x_t)_{\hat{x}})}{\min(s_\theta^p(x_t)_{\hat{x}}, s^p(x_t)_{\hat{x}})} - 1 \right) \right| + \epsilon_p$$

$$\leq C_2 \left| \left( \frac{\max(s_\theta^p(x_t)_{\hat{x}}, s^p(x_t)_{\hat{x}}) - \min(s_\theta^p(x_t)_{\hat{x}}, s^p(x_t)_{\hat{x}})}{\min(s_\theta^p(x_t)_{\hat{x}}, s^p(x_t)_{\hat{x}})} \right) \right| + \epsilon_p$$

$$\leq \frac{C_2}{\min(s_\theta^p(x_t)_{\hat{x}}, s^p(x_t)_{\hat{x}})} \left| (s_\theta^p(x_t)_{\hat{x}} - s^p(x_t)_{\hat{x}}) \right| + \epsilon_p$$

$$\leq \frac{C_2}{C_1} \left| (s_\theta^p(x_t)_{\hat{x}} - s^p(x_t)_{\hat{x}}) \right| + \epsilon_p$$

$$\leq \frac{C_2}{C_1} \epsilon_p + \epsilon_p$$

$$= \epsilon_p\left(1 + \frac{C_2}{C_1}\right)$$

Then, we continue with the second term of Equation (12), which is simply

$$\left| s_\phi^q(x_t)_{\hat{x}} - s^q(x_t)_{\hat{x}} \right| \leq \epsilon_q$$

Finally, we write the term of Equation (12)

$$\left| s_\theta^p(x_t)_{\hat{x}} \log\big(s_\phi^q(x_t)_{\hat{x}}\big) - s^p(x_t)_{\hat{x}} \log\big(s^q(x_t)_{\hat{x}}\big) \right|$$

$$\leq \max\big(s_\theta^p(x_t)_{\hat{x}}, s^p(x_t)_{\hat{x}}\big) \left| \log s^q(x_t)_{\hat{x}} - \log\big(s_\phi^q(x_t)_{\hat{x}}\big) \right|$$

$$\leq C_2 \left| \log s^q(x_t)_{\hat{x}} - \log\big(s_\phi^q(x_t)_{\hat{x}}\big) \right|$$

$$\leq \frac{C_2}{C_1} \left| s^q(x_t)_{\hat{x}} - s_\phi^q(x_t)_{\hat{x}} \right|$$

$$\leq \frac{C_2}{C_1} \epsilon_q$$

Combining the results we obtain

$$\left| \mathbb{E}_{x \sim p} \mathcal{E}(s_\theta^p, s_\phi^q; x) - \mathbb{E}_{x \sim p} \mathcal{E}(s^p, s^q; x) \right|$$

$$\leq \left| \int_0^T \mathbb{E}_{x_t \sim \overrightarrow{p}_{t|0}(\cdot|x_0)} \sum_{\hat{x} \neq x_t} \overrightarrow{Q}(x_t, \hat{x}) \left( (\epsilon_p + \epsilon_q)\left(1 + \frac{C_2}{C_1}\right) \right) dt \right|$$

$$\leq \bar{\sigma}(T) D|\chi| \left( (\epsilon_p + \epsilon_q)\left(1 + \frac{C_2}{C_1}\right) \right)$$

where the factor $D|\chi|$ is the number of non-zero entries in $\overrightarrow{Q}$ using the absorb configuration. We can finally put together everything to get the error bound for Equation (11)

$$\left| \mathbb{E}_{x \sim p} \mathcal{E}(s_\theta^p, s_\phi^q; x) - \text{KL}\left[ \overrightarrow{p}_T \,\|\, \overrightarrow{q}_T \right] - \mathbb{E}_{x \sim p} \mathcal{E}(s^p, s^q; x) \right|$$

$$\leq \left| \mathbb{E}_{x \sim p} \mathcal{E}(s_\theta^p, s_\phi^q; x) - \mathbb{E}_{x \sim p} \mathcal{E}(s^p, s^q; x) \right| + \text{KL}\left[ \overrightarrow{p}_T \,\|\, \overrightarrow{q}_T \right]$$

$$\leq \bar{\sigma}(T) D|\chi| \left( (\epsilon_p + \epsilon_q)\left(1 + \frac{C_2}{C_1}\right) \right) + (1 - \overrightarrow{p}_T(\emptyset^D))(D \log |\chi| C_2)$$

Note that $\overrightarrow{p}_T(\emptyset^D)$ approaches 1 exponentially in $T$ (Equation 14 of Lou et al. (2024)). The other term, instead, shows that the error on the KL increases linearly with the error of the score network for fixed $T$ and does not depend on terms exponential on the KL divergence.

### E.3 DISCUSSION ON CONSISTENCY

As presented in Appendix E, the error of INFO-SEDD decomposes in two terms: one proportional to the error of the score networks, and one proportional to $\overrightarrow{p}_T(\emptyset^D)$.

$$\left| \mathbb{E}_{x\sim p}\mathcal{E}(s_\theta^p, s_\phi^q; x) - \text{KL}\left[\overrightarrow{p}_T \| \overrightarrow{q}_T\right] - \mathbb{E}_{x\sim p}\mathcal{E}\left(s^p, s^q; x\right) \right|$$

$$\leq \left| \mathbb{E}_{x\sim p}\mathcal{E}(s_\theta^p, s_\phi^q; x) - \mathbb{E}_{x\sim p}\mathcal{E}\left(s^p, s^q; x\right) \right| + \text{KL}\left[\overrightarrow{p}_T \| \overrightarrow{q}_T\right]$$

$$\leq \bar{\sigma}(T)D|\chi|\left((\epsilon_p + \epsilon_q)\left(1 + \frac{C_2}{C_1}\right)\right) + (1 - \overrightarrow{p}_T(\emptyset^D))(D\log|\chi|C_2) \tag{13}$$

The second term of Equation (13) cannot be reduced with better score networks, as it only depends on the CTMC and the data distribution. However, it decreases exponentially in $\bar{\sigma}(T)$ (Equation 14 of Lou et al. (2024)), establishing a trade-off with the first term, which only increases linearly with $\bar{\sigma}(T)$. For this reason, we focus on the first term. Note that our final KL estimator Equation (4) is a continuous functional of the score networks $s_\theta^p, s_\phi^q$. By theorem 3.4 of Lou et al. (2024), the DWDSE loss is an equivalent objective compared to the score entropy loss. This allows us to apply proposition 3.2 of Lou et al. (2024), for which, given enough samples and model capacity, the score network $s_\theta^p$ and $s_\phi^q$ converge to their true values $s^p$ and $s^q$. Our final KL estimator Equation (4) is a continuous functional of these score networks. By the Continuous Mapping Theorem (theorem 1.9.5 of Vaart and Wellner (1997)), assuming as in Lou et al. (2024) that $s_\theta^p$ and $s_\phi^q$ converge in probability to their true values $s^p$ and $s^q$, the output of the functional (our KL estimate) must also converge in probability to the true KL divergence minus the truncation bias.

### E.4 DISCUSSION ON BIAS

Consider the INFO-SEDD estimator

$$\alpha = \mathbb{E}\left[\int_0^T \sum_{x\neq\overrightarrow{X}_t} \overrightarrow{Q}_t(\overrightarrow{X}_t, x)\left(K\left(s^p(\overrightarrow{X}_t)_x\right) + s^q(\overrightarrow{X}_t)_x - s^p(\overrightarrow{X}_t)_x \log s^q(\overrightarrow{X}_t)_x\right)dt\right]$$

and its empirical version over a collection of samples $x_0^i \sim p_{data}$

$$\hat{\alpha} = \frac{1}{N}\sum_{i=1}^N\left[\int_0^T \sum_{x\neq x_t^i} \overrightarrow{Q}_t(x_t^i, x)\left(K\left(s^p(x_t^i)_x\right) + s^q(x_t^i)_x - s^p(x_t^i)_x \log s^q(x_t^i)_x\right)dt\right].$$

Let us denote $\phi(x_t) = \sum_{x\neq x_t^i} \overrightarrow{Q}_t(x_t^i, x)\left(K\left(s^p(x_t^i)_x\right) + s^q(x_t^i)_x - s^p(x_t^i)_x \log s^q(x_t^i)_x\right)$, then we can write its expectation as

$$\mathbb{E}[\hat{\alpha}] = \mathbb{E}\left[\frac{1}{N}\sum_{i=1}^N \int_0^T \phi(x_t^i)dt\right]$$

$$= \frac{1}{N}\sum_{i=1}^N \mathbb{E}\left[\int_0^T \phi(x_t^i)dt\right]$$

Note that $\mathbb{E}\left[\int_0^T \phi(x_t^i)dt\right]$ is the true $\alpha$. As a consequence $\mathbb{E}[\hat{\alpha}] = \alpha$, which means that INFO-SEDD is an unbiased estimator for $\alpha$. Note, however, that we still have the truncation bias which is bounded by

$$|\text{KL}\left[p_{data} \| q_{data}\right] - \alpha| \leq (1 - \overrightarrow{p}_T(\emptyset^D))(D\log|\chi|C_2).$$

This source of bias is object of tradeoff with the other term

$$\bar{\sigma}(T)D|\chi|\left((\epsilon_p + \epsilon_q)\left(1 + \frac{C_2}{C_1}\right)\right)$$

since increasing $T$ would reduce the former but expand the latter. However, the term $(1 - \vec{p}_T(\emptyset^D))$ decreases exponentially in $\bar{\sigma}(T)$ (Equation 14 of Lou et al. (2024)). This means that the truncation bias can be drastically reduced without affecting the other error term which only has a linear dependence on $\bar{\sigma}(T)$.

