# OpenReview forum: "Information Estimation with Discrete Diffusion"
_ICLR.cc/2026/Conference — ICLR 2026 Poster_

### Official Review · Reviewer_gDGn · 2025-10-21

**Soundness:** 3
**Presentation:** 2
**Contribution:** 2
**Rating:** 4
**Confidence:** 3

**Summary:**

The paper focuses on mutual information for high-dimensional discrete data. The score function of a diffusion model is used as a key component to estimate mutual information and entropy from a trained diffusion model. In comparison, prior work on diffusion models for mutual information estimation focuses on continuous random variables. The proposed method INFO-SEDD is implemented in two variants, one that approximates MI as the KL divergence between the product distribution and the marginals and one that compares the conditional $p_{Y|X}$ to $p_Y$. The approaches are benchmarked on synthetic discrete data pairs with high dimensions. Further, the authors consider case studies for text and DNA sequence data.

**Strengths:**

- The authors consider a rich set of benchmark tasks and potential real-world applications.
- The performance of the proposed method compared to prior work on continuous data seems promising.
- Connecting the continuous time Markov chains and diffusion models to compute MI for discrete data seems new to the reviewer.

**Weaknesses:**

- The paper is very dense and hard to follow since many parts are pushed to the appendix (see questions and comments below).
- It is not clear what the exact differences to other diffusion-based MI estimators are, e.g., MINDE, and how the authors manage to model discrete distributions compared to prior work that did not cover this case.
- Experiments:
	- It is not clear how the baselines are implemented for Sec. 4.1. Don’t they all focus on continuous data? As a simple baseline, using the plug-in estimator for discrete MI, and some more elaborate simple estimators (e.g., Goebel et al. (2005)) would improve the evaluation.
	- Real-world examples: The examples on text summarization and genomics are interesting, but they do not allow clear inferences about the performance of the MI estimator. The consistency test on the genomics dataset seems to not go beyond classification.
	- Typically, papers on discrete MI discuss the issue of bias in the plug-in estimator, which occurs under high dimensionality (in comparison to the sample size). It would be interesting to see if the proposed estimator can effectively mitigate such issues (see, e.g., Goebel et al. (2005)). The authors do evaluate susceptibility to dimensionality, but adopt very large sample sizes ($10^6$) for D=256 and 1024 and employ a model with 2M parameters. Here again, a comparison to the plug-in estimator and improvements of it to high-dimensions should be compared to.

Minor points:
- To which data is the GP fitted (line 356)?
- What is meant by “simple and uninteresting scenarios” (line 124/125)?

Reference:
- Goebel et al. (2005). An approximation to the distribution of finite sample size mutual information estimates.

**Questions:**

- The training of the method is not clear. Do the authors use a pre-trained diffusion model?
- The authors state that MI can be used for model selection. How does it compare to other metrics like the model perplexity?

---

> ### Author Response · Authors · 2025-11-20
>
> Thank you for your careful review of our work.
> We really appreciate the positive feedback on the novelty of our method and that you value the extent of our experimental protocols and promising results.
> We address your very useful comments and questions below, which were helpful in further improving our paper.
>
> 1. **Differences with MINDE**.
> The difference between Infosedd and MINDE resides in the distinct nature of the data distributions that they consider. MINDE focuses on distributions supported on the manifold $R^N$, in which the notion of geometric distance is meaningful and for which it is possible to define "gradients" of a log-density (i.e., the time varying score functions of [1]). Infosedd addresses the problem of estimating information metrics in spaces where the notion of geometry is less meaningful. Thus, it defines the discrete score, which compensates for the lack of a proper distance metric by considering all possible neighbor pairs. A link between the two approaches exists: discrete and continuous scores are connected, as discussed in Proposition 1, in [2].
> Finally, note that the so called "embedding trick" is effectively a way to induce an appropriate geometry on categorical data. In principle, this is a valid option to adapt methods designed for continuous distributions to work on discrete distributions.
>
> [1] Song et al. Score-based generative modeling through stochastic differential equations, International Conference on Learning Representations, 2021
>
> [2] Meng et al. Concrete Score Matching: Generalized Score Matching for Discrete Data, Neural Information Processing Systems, 2022
>
> 2. **How did you implement the baselines**? Most of the available competitors focus on continuous data. The research gap in MI estimators for discrete data is the main motivation behind Infosedd. As stated in our paper, the typical workaround to use existing methods designed for continuous data is the "embedding trick," which involves projecting discrete data into a continuous embedding space. Our whole experimental campaign is geared toward answering the question of whether such a workaround is sufficient, or if a novel MI estimator specifically designed for discrete data would be a better choice. Our results indicate that an appropriate method for discrete data outperforms the alternatives.
>
> We also fully agree with the reviewer that including plug-in estimators in our evaluation protocol would strengthen our work. Note that the estimator by [3]. does not seem to be suitable for large MI estimation (at page 3 of their paper, the authors state that MI should be lower than 0.2 bits). For this reason, we benchmarked a recent plug-in estimator proposed in [4], focusing on high-MI cases from Table 1. We report the results below:
>
> | Ground truth MI | Estimate |
> | :--- | :--- |
> | 10 | 11.56 |
> | 20 | 9.33 |
> | 30 | 10.55 |
> | 40 | 2.87 |
> | 50 | 4.18 |
>
> Our results indicate that plug-in MI estimators might encounter difficulties in high-MI scenarios, as their performance degrades with higher MI values. This further motivates the need for our proposed method.
>
> [3] Goebel et al. An approximation to the distribution of finite sample size mutual information estimates, IEEE International Conference on Communications, 2005
>
> [4] Gao et al. Estimating mutual information for discrete-continuous mixtures, Neural Information processing Systems, 2017
>
> 3. **Real world examples are not sufficient to measure performance**. We agree with the reviewer: real-world data do not offer ground-truth MI, so it is challenging to compare alternative methods. This is the reason why we decided to split our evaluation protocol according to 1) a synthetic benchmark in which performance can be measured reliably; 2) real-world examples for which we derived an expected behavior in the absence of ground-truth information. In both cases, Infosedd outperforms existing methods. Additionally, Appendix C covers another practical scenario: we use Infosedd to estimate the entropy of Ising models, for which it is possible to obtain ground-truth values.
> Overall, our goal in this work is to demonstrate that Infosedd is a reliable MI estimator and that there are many practical endeavors that can benefit from accurate information measure estimation.

---

> ### Author Response · Authors · 2025-11-20
>
> 4. **Consistency tests vs. classification**. We agree with the reviewer, specifically in the context of the Genomics benchmark (first part of Section 4.3). Our consistency tests show that Infosedd is a sensible alternative to classification for such tasks. More broadly, traditional genomic studies rely on simple linear correlation, as prevalent in Genome Wide Association Studies [5], and classification models [6]. Recently, the genomics community has advocated for using information-theoretic measures [7] as valid successors to current practice.
> Finally, as noted by reviewer Rjnfn, the ability of Infosedd to perform amortized inference on subsequences is very useful, as shown in the experiments on Motif detection (second part of Section 4.3), which is a harder task compared to classification.
> Overall, our experiments position Infosedd as a reliable and versatile tool for solving a variety of problems in genomics.
>
> [5] Uffelman et al. Genome-wide association studies, Nature Review Methods Primers, 2021
>
> [6] Huang et al. Twenty years of genome-wide association studies: Health translation challenges and AI opportunities, European Journal of Human Genetics, 2025
>
> [7] Galas et al. Toward an information theory of quantitative genetics, Journal of Computational Biology, 2021
>
> 5. **Estimation bias.** We agree that additional discussions on consistency and bias are important and confer additional solidity to our proposed estimator.
> We prepared additional, detailed discussions on consistency and bias, building on our error analysis from Appendix E.
> Using the error bound at the end of Appendix E, we can identify two sources of error: 1) a term that is linearly dependent on the score network error, 2) the truncation error, which is a positive term that depends solely on the data distribution and the CTMC.
> The second term ($\text{KL}[p_T \|\| q_T]$) is in the form of $(1 - p_T(\emptyset^D))(D \log |\chi| C_2)$. Our analysis establishes a tradeoff with the error term due to the score. While the score error increases linearly with $\bar{\sigma}(T)$, the truncation error decreases exponentially with $\bar{\sigma}(T)$ (see also Equation 14 of [8]). In practice, using standard noise schedules, the exponential dependence on $\bar{\sigma}(T)$ makes the second term negligible.
>
> Building on these results, we extend our analysis about consistency and bias in a new Appendix F.
>
> Concerning consistency (Appendix F.1), we adopt the same strategy as in [8], proposition 3.2, where it is discussed how, given enough samples and model capacity, the score networks converge to their true values. Then, since the term in Equation (5) is a continuous functional (which depends on score networks, not true score functions), it similarly converges to its true value. To prove this, it is necessary to leverage the Continuous Mapping Theorem (theorem 1.9.5 of [9]), which demonstrates that consistency is preserved by continuous mappings.
>
> Concerning bias (Appendix F.2), we note that it can arise from using a finite number of MonteCarlo samples to compute numerically the expectations, which is standard practice in neural estimators. In general, as discussed in [10], variational estimators of divergences are intrinsically biased at any finite sample size used for the MC integrations. These errors arise from the fact that, e.g., $E_{X_i}[\log 1/N_{MC} \sum_{i=1}^{N_{MC}} f(X_i)] \neq \log E_{X}[f(X)]$. On the other hand, Infosedd does not suffer from the same limitations. Clearly, a complete analysis would require studying how the score approximation error scales as a function of the training set size. While recent results in the literature are available (see e.g. [11]), such analysis is limited to a simplistic class of neural networks and falls outside the scope of our work. Nevertheless, we remind the reviewer that we have included an empirical analysis of the scaling of the error as a function of the number of samples in Appendix C.1.5
>
> [8] Lou et al. Discrete diffusion modeling by estimating the ratios of the data distribution, International Conference on Machine Learning,  2024
>
> [9] Vaart et al. Weak convergence and empirical processes: with applications to statistics, Springer Sciences, 1997
>
> [10] Song et al. Understanding the limitations of vaiational mutual information estimators, International Conference on Learning Representations, 2020
>
> [11] Srikanth et al. Discrete State Diffusion Models: A Sample Complexity Perspective, arXiv preprint 2025

---

> ### Author Response · Authors · 2025-11-20
>
> 6. **Plugin estimator extreme dimensionality** Again, we agree with the reviewer that adding a plug-in baseline to the extreme dimensionality experiments would improve the paper. We benchmark the recent plug-in estimator proposed in [4] on the extreme dimensionality benchmark, and obtain the following results:
>
> | Ground truth MI | D | Estimate |
> | :--- | :--- | :--- |
> | 0.5 | 256 | 14.17 |
> | 0.5 | 1024 | 14.32 |
>
> These findings help us reinforce our claim that a neural estimator for discrete data is truly needed. However, we agree with the reviewer that plug-in estimators are a useful tool in the right circumstances.
>
> [4] Gao et al. Estimating mutual information for discrete-continuous mixtures, Neural Information processing Systems, 2017
>
> 7. **To which data is the GP fitted (line 356)**? The data is fitted using the Mutual Information estimates from infosedd and the consistency human metric.
>
> 8. **What is meant by simple and uninteresting scenarios (line 124/125)**? We refer to scenarios in which the closed form of the data distribution is available. In our revision, we will modify the text and remove the "uninteresting" adjective.
>
> 9. **The training of the method is not clear. Do the authors use a pre-trained diffusion model?**
> In general, Infosedd **does not require** access to a pretrained diffusion model. The training objective is valid for any dataset for which no pretrained models is available. However, Infosedd **can leverage** pretrained models that are available for specific domains, which is a strength of our approach. Indeed, as clarified in our main paper, in the text experiments (Section 4.2) we use MDLM small, a pretrained masked diffusion model, as a backbone. In the genomics experiments (Section 4.3) we use the Caduceus-1K model, a pretrained masked language model. For the experiments on synthetic data (Section 4.1), we train our models from scratch, using standard practice [8].
>
> [8] Lou et al. Discrete diffusion modeling by estimating the ratios of the data distribution, International Conference on Machine Learning,  2024
>
> 10. **Can you do model selection using perplexity**? We are not aware of prior work using perplexity as a metric for model selection in text summarization. However, in appendix G.3, we provide correlation matrices that include Infosedd, human metrics, as well as the BERTSCORE [12], BLEU [13] and CIDER [14] metrics, which are very popular in the text summarization community. In particular, we can see from figure 11 and figure 12 in appendix G.3 that Infosedd-c correlates better than BERTSCORE, BLEU and CIDER on the consistency human metric. Additionally, infosedd creates a lower barrier to entry for model selection, as it is reference-free and does not require gold-standard summaries.
>
> [12] Zhang et al. Bertscore: Evaluating text generation with bert, International Conference on Learning Representations, 2020
>
> [13] Papineni et al. Bleu: a method for automatic evaluation of machine translation, Proceedings of the 40th Annual Meeting of the Association for Computational Linguistics, 2002
>
> [14] Vedantam et al. Cider: Consensus-based image description evaluation, IEEE Conference on Computer Vision and Pattern Recognition, 2005

---

> > ### Comment · Reviewer_gDGn · 2025-11-24
> >
> > Thank you very much for the replies and clarifications. I have a brief follow-up question to the response to point 1 and 2:
> >
> > It is still not completely clear to me which concrete adjustments need to be made in practice. Do we only need an embedding layer and can then apply MINDE? If not, is this what is done for MINDE as a baseline?

---

> > > ### Author Response · Authors · 2025-11-24
> > >
> > > Dear reviewer, we thank you for the follow-up question. Indeed, concerning MINDE and other estimators conceived for continuous distributions, we use an embedding layer that projects discrete data into a continuous space. However, our results consistently indicate that this approach is suboptimal compared to a native discrete method, reinforcing the need for Infosedd.

---

### Official Review · Reviewer_ps2M · 2025-10-24

**Soundness:** 3
**Presentation:** 2
**Contribution:** 3
**Rating:** 6
**Confidence:** 3

**Summary:**

The authors propose a discrete diffusion-based approach that mitigates the problem of continuous space MI estimators in high data dimensionality. The proposed approach builds upon Dynkin’s lemma and can be used to estimate mutual information or entropy. Numerical experiments validate the proposed estimator compared to other state-of-the-art estimators. The authors design consistency tests for real-world textual and genomics data.

**Strengths:**

- Good theoretical contributions (including Appendix E)
- Promising experimental results
- The proposed approach differs from many continuous existing MI estimators
- The proposed estimator appear to be converging faster than other sota estimators

**Weaknesses:**

- The paper is not easy to read
- In line 200 you write that expressing the MI as the KL divergence requires training two score models. However, many MI estimators target the estimation of the density ratio. So this sentence should be rewritten.
- Line 1211 typo: $\mathcal{X}|$
- The code is not provided

**Questions:**

- What is $\delta()$ in equation 1? It is not defined
- Table 1 does not report standard deviation information. Can the authors provide some results that also quantify the variance of the proposed estimator compared to the others?
- The authors claim that INFO-SEDD is lightweight (line 21), but I do not see any result reporting computational times in the main part of the paper. Can the authors provide a table with the computational time comparison of the different estimators? In Appendix C.1.3 the authors report the runtime for one experiment, but it would be meaningful to see these results in the main part of the paper and summarized over multiple scenarios. The authors report that INFO-SEDD uses less memory than SMILE and GAN-DIME, so is “lightweight” referred to this?
- In Appendix C.1.4, Table 3 shows that MINDE overestimates MI for low values. Is this result generalizable for different scenarios? Could you claim that this happens to many generative MI estimators except INFO-SEDD? Or is it just a coincidence?
- How do the other methods perform in the extreme dimensionality experiments in C.1.7?
- In Table 5, increasing $|\mathcal{X}|$ should lead to a better performance of the other estimators, right? Why does the trend appear to be opposite?
- Why did the authors decide to not provide the code?

---

> ### Author Response · Authors · 2025-11-20
>
> We thank the reviewer for carefully reviewing our work, including a detailed analysis of the Appendices.
> We really appreciate the depth and  positive remarks about our contribution, its properties, and its practical utility.
> Comments and questions, which we address below, are helpful and contribute to tangible improvements in our paper.
>
> 1. **The paper is not easy to read**. Thank you for spotting typos and unclear parts, which we will fix.
>
> 2. **Expression for the KL divergence**. We are aware that some work in the literature computes MI by estimating the density ratio. In our sentence, we refer to a practical concern when estimating MI using two score models, as in Equation (5). To overcome this limitation, we introduce masked tokens as a way to use a single score model. We will pay attention to clarifying the scope of our sentence in the main text.
>
> 3. **Missing source code**. In our original submission, we have provided the code at the link https://anonymous.4open.science/r/mutinfo-diffusion-860B, as visible in the reproducibility statement. We will link our complete repository, with example scripts and instructions, for the camera ready version of the paper.
>
> 4. **What is $\(\delta()\)$**? This corresponds to the Dirac delta.
>
> 5. **Missing std deviation in Table 1**. We agree with the reviewer, and aim to corroborate results with statistical confidence. We thus propose an updated version of the table, obtained by running multiple experiments for 10 different seeds, and report standard deviation:
>
> |  | infosedd | gan-dime | hd-dime | kl-dime | minde | mine | nwj | smile |
> | --- | --- | --- | --- | --- | --- | --- | --- | --- |
> | $MI=10, D=10$ | 9.92 $\pm$ 0.12 | 12.15 $\pm$ 0.89 | 9.73 $\pm$ 0.43 | 8.38 $\pm$ 0.90 | 14.01 $\pm$ 2.91 | 10.21 $\pm$ 6.33 | 6.16 $\pm$ 2.11 | 12.83 $\pm$ 0.95 |
> | $MI=20, D=20$ | 20.02 $\pm$ 0.21 | 22.09 $\pm$ 1.75 | 12.65 $\pm$ 1.07 | 7.51 $\pm$ 0.56 | 26.98 $\pm$ 3.16 | 8.82 $\pm$ 0.80 | 6.50 $\pm$ 0.84 | 23.11 $\pm$ 1.41 |
> | $MI=30, D=30$ | 29.83 $\pm$ 0.54 | 20.74 $\pm$ 1.75 | 11.72 $\pm$ 2.69 | 7.02 $\pm$ 0.43 | 31.08 $\pm$ 4.33 | 7.41 $\pm$ 1.23 | 6.35 $\pm$ 0.34 | 21.79 $\pm$ 1.08 |
> | $MI=40, D=40$ | 39.11 $\pm$ 0.65 | 19.64 $\pm$ 1.33 | 11.68 $\pm$ 0.94 | 6.52 $\pm$ 0.32 | 33.97 $\pm$ 3.32 | 6.91 $\pm$ 0.66 | 6.24 $\pm$ 0.59 | 20.13 $\pm$ 1.27 |
> | $MI=50, D=50$ | 47.77 $\pm$ 1.18 | 17.27 $\pm$ 1.46 | 10.47 $\pm$ 1.12 | 6.41 $\pm$ 0.62 | 32.60 $\pm$ 3.93 | 7.21 $\pm$ 1.14 | 5.95 $\pm$ 0.31 | 18.97 $\pm$ 1.05 |
>
> Our updated results confirm that infosedd outperforms all alternative methods.
> We observe that, in addition to improved accuracy, infosedd also has lower and more stable variance compared to other baselines, which is a highly desirable property. Indeed, variational estimators suffer from a variance that scales with the ground-truth MI value (see theorem 2 in [1]).
>
> [1] Song et al. Understanding the limitations of vaiational mutual information estimators, International Conference on Learning Representations, 2020
>
> 6. **Computational efficiency**. We understand that computational efficiency is crucial. For this reason, we contextualize the results of Table 1, expanding the discussion in appendix C1.3 by providing an analysis of memory and runtime requirements. We report the peak memory usage over one epoch (in MB):
> |  | infosedd | gan-dime | hd-dime | kl-dime | minde | mine | nwj | smile |
> | --- | --- | --- | --- | --- | --- | --- | --- | --- |
> | $MI=10, D=10$ | 346.37 | 645.81 | 645.81 | 645.81 | 440.51 | 645.81 | 645.81 | 645.81 |
> | $MI=20, D=20$ | 687.22 | 1301.98 | 1301.98 | 1301.98 | 876.44 | 1301.98 | 1301.98 | 1301.98 |
> | $MI=30, D=30$ | 1005.66 | 1913.15 | 1913.15 | 1913.15 | 1286.35 | 1913.15 | 1913.15 | 1913.15 |
> | $MI=40, D=40$ | 1345.94 | 2564.33 | 2564.33 | 2564.33 | 1726.25 | 2564.32 | 2564.32 | 2564.32 |
> | $MI=50, D=50$ | 1666.30 | 3180.50 | 3180.50 | 3180.50 | 2138.41 | 3180.49 | 3180.49 | 3180.49 |
>
> and runtime of one epoch (in seconds):
> |  | infosedd | gan-dime | hd-dime | kl-dime | minde | mine | nwj | smile |
> | --- | --- | --- | --- | --- | --- | --- | --- | --- |
> | $MI=10, D=10$ | 2.40 | 2.89 | 2.90 | 2.86 | 2.16 | 2.89 | 2.96 | 2.94 |
> | $MI=20, D=20$ | 3.34 | 3.99 | 4.02 | 3.98 | 2.62 | 4.03 | 4.03 | 4.00 |
> | $MI=30, D=30$ | 4.51 | 5.56 | 5.54 | 5.57 | 3.37 | 5.57 | 5.57 | 5.61 |
> | $MI=40, D=40$ | 5.96 | 7.34 | 7.33 | 7.36 | 4.00 | 7.39 | 7.42 | 7.37 |
> | $MI=50, D=50$ | 7.06 | 8.71 | 8.69 | 8.71 | 4.68 | 8.76 | 8.73 | 8.72 |
>
> As we can see, Infosedd requires around 20% less time to run and around 50% less memory compared to variational competitors, offering a better performance-accuracy trade-off for applications like sufficient statistics learning that require frequent MI estimates. Indeed, the adjective *lightweight* in the abstract refers to the lighter runtime and memory requirements compared to other estimators.

---

> > ### Comment · Reviewer_ps2M · 2025-11-24
> >
> > I am satisfied with the authors reply. I will keep my positive score and increase the confidence.

---

> > > ### Author Response · Authors · 2025-11-24
> > >
> > > Dear reviewer, thank you for acknowledging our reply. We are glad it helped conserving your positive judgement. We hope this will contribute to a positive outcome for our work.

---

> ### Author Response · Authors · 2025-11-20
>
> 7. **MINDE overestimates**. This is an interesting remark! However, our current results do not grant us the right to claim that such behavior is general and that it applies to other generative MI estimators. Indeed, the class of generative MI estimation methods is very broad [2, 3], which calls for an individual, case-by-case analysis, to confirm a general behavior.
>
> [2] Butakov et al. Mutual information estimation via normalizing flows, Neural Information Processing Systems 2024
>
> [3] Wang et al. Information Theoretic Text-to-Image Alignment, International Conference on Learning Representations 2025
>
>
> 8. **Additional extreme dimensionality experiments**. To complement our results in Appendix C.1.7, we now benchmark GAN-DIME, HD-DIME and SMILE in high-dimensionality, for the case ground truth MI=0.5. These methods perform relatively well for the D=256 case, with the following estimates:
> - HD-DIME: 0.43
> - SMILE: 0.67
> - GAN-DIME: 0.51
>
> However, when pushing to the case D=1024, all of these methods fail:
> - HD-DIME: 0.00
> - SMILE: 0.00
> - GAN-DIME: 0.00
>
> We thus have even more compelling evidence about the performance of Infosedd in such extreme scenarios.
>
> 9. **Table 5 results**. As the support dimension $|\chi|$ increases, the probability distribution becomes more complex. The experience we gained from our experimental results leads us to believe that it is natural for MI estimators to struggle in higher dimensional cases compared to lower dimensional ones. Other works, albeit in the continuous domain, found similar patterns (Table 1 in [4]): high-dimensional data is a painpoint for many existing MI estimators.
>
> [4]: Franzese et al. MINDE: Mutual information neural diffusion estimation, International Conference on Learning Representations 2024

---

### Official Review · Reviewer_C1dg · 2025-11-01

**Soundness:** 3
**Presentation:** 3
**Contribution:** 3
**Rating:** 6
**Confidence:** 4

**Summary:**

The paper proposes a new algorithm for estimating the KL divergence (and consequently, entropy and mutual information) in high-dimensional discrete distributions. The method expresses the KL divergence as a conditional expectation and applies Dynkin’s formula to derive a KL estimator that depends on two density ratios. Each density ratio is then approximated using a diffusion-weighted denoising score entropy loss. The method is evaluated across a range of mutual information and entropy estimation tasks.

**Strengths:**

The proposed method for estimating mutual information in discrete data is novel. The reviewer is not aware of any previous works that apply discrete (CTMC-based) diffusion to the mutual information estimation task.
The method shows potential practical utility. The results on estimating the entropy of Ising models are promising, and the reviewer believes the approach could be applied to study phase transitions in more complex statistical physics systems with discrete microstate space. As another practical utility one can apply the proposed method to learning neural sufficient statistics and representation learning.

**Weaknesses:**

It is unclear whether the proposed mutual information estimator is consistent and unbiased. Additionally, although the error analysis is included in the supplementary material, it is not discussed in the main text.

In general, the theoretical contribution of the proposed method is limited and primarily based on a reformulation of the results by Lou et al [1]. For example, the proposed KL estimator has the same form as the diffusion-weighted denoising score entropy (Equation 10 in Lou et al.), with the main difference being the replacement of the density ratio. Moreover, the use of Dynkin’s formula was already presented in Lou et al. (Theorem 3.6).

**Questions:**

Could you provide a proof that the proposed estimator is consistent and unbiased? For example using a similar strategy as the proof of consistency of the score entropy loss in Lou et. al. (Proposition 3.2).

What is the computational complexity of the proposed estimator?
Regarding the broader impact, the paper would benefit from a discussion on sufficient statistics learning (e.g., see [2]). Including a computational complexity analysis would clarify whether the method can be feasibly used as a loss function in downstream tasks.

[1] Lou et. al. Discrete Diffusion Modeling by Estimating the Ratios of the Data Distribution, 2020.

[2] Y Chen et. al. Neural Approximate Sufficient Statistics for Implicit Models, 2020.

---

> ### Author Response · Authors · 2025-11-20
>
> We thank the reviewer for the careful assessment of our work.
> We are glad that the reviewer finds our work original and sound. We also thank the reviewer for spending time on our appendices finding applications beyond what was reported in the main paper (e.g. Ising models).
> The reviewer raises important questions, which we address below:
>
> 1. **Theory not discussed in the main text**. We agree that the theoretical part of the paper deserves at least a discussion in the main paper. Moreover, we agree that additional discussions on consistency and bias are important and confer additional solidity to our proposed estimator.
> We prepared additional, detailed discussions on consistency and bias, building on our error analysis from Appendix E.
> Using the error bound at the end of Appendix E, we can identify two sources of error: 1) a term that is linearly dependent on the score network error, 2) the truncation error, which is a positive term that depends solely on the data distribution and the CTMC.
> The second term ($\text{KL}[p_T \|\| q_T]$) is in the form of $(1 - p_T(\emptyset^D))(D \log |\chi| C_2)$. Our analysis establishes a tradeoff with the error term due to the score. While the score error increases linearly with $\bar{\sigma}(T)$, the truncation error decreases exponentially with $\bar{\sigma}(T)$ (see also Equation 14 of [1]). In practice, using standard noise schedules, the exponential dependence on $\bar{\sigma}(T)$ makes the second term negligible.
>
> Building on these results, we extend our analysis about consistency and bias in a new Appendix F.
>
> Concerning consistency (Appendix F.1), we adopt the same strategy as in [1], proposition 3.2, where it is discussed how, given enough samples and model capacity, the score networks converge to their true values. Then, since the term in Equation (5) is a continuous functional (which depends on score networks, not true score functions), it similarly converges to its true value. To prove this, it is necessary to leverage the Continuous Mapping Theorem (theorem 1.9.5 of [2]), which demonstrates that consistency is preserved by continuous mappings.
>
> Concerning bias (Appendix F.2), we note that it can arise from using a finite number of MonteCarlo samples to compute numerically the expectations, which is standard practice in neural estimators. In general, as discussed in [3], variational estimators of divergences are intrinsically biased at any finite sample size used for the MC integrations. These errors arise from the fact that, e.g., $E_{X_i}[\log 1/N_{MC} \sum_{i=1}^{N_{MC}} f(X_i)] \neq \log E_{X}[f(X)]$. On the other hand, Infosedd does not suffer from the same limitations. Clearly, a complete analysis would require studying how the score approximation error scales as a function of the training set size. While recent results in the literature are available (see e.g. [4]), such analysis is limited to a simplistic class of neural networks and falls outside the scope of our work. Nevertheless, we remind the reviewer that we have included an empirical analysis of the scaling of the error as a function of the number of samples in Appendix C.1.5
>
> [1] Lou et al. Discrete diffusion modeling by estimating the ratios of the data distribution, International Conference on Machine Learning,  2024
>
> [2] Vaart et al. Weak convergence and empirical processes: with applications to statistics, Springer Sciences, 1997
>
> [3] Song et al. Understanding the limitations of vaiational mutual information estimators, International Conference on Learning Representations, 2020
>
> [4] Srikanth et al. Discrete State Diffusion Models: A Sample Complexity Perspective, arXiv preprint 2025
>
>
> 2. **Limited theoretical contribution**. Indeed, Infosedd relies on methods from [1], but focuses on KL estimation, which unlocks the ability to study mutual information and entropy measures. This key connection is what allows us to build upon prior work on generative modeling and recast it for information measure estimation. We believe this to be a strength of our proposed method, rather than a weakness, and it is also the main reason why, for the narrative of our work, we decided to emphasize experimental results and applications, deferring theoretical analysis to the appendix.
>
> [1] Lou et al. Discrete diffusion modeling by estimating the ratios of the data distribution, International Conference on Machine Learning,  2024

---

> ### Author Response · Authors · 2025-11-20
>
> 3. **Computational complexity**.
> We understand that computational efficiency is crucial. For this reason, we contextualize the results of Table 1, expanding the discussion in appendix C1.3 by providing an analysis of memory and runtime requirements. We report the peak memory usage over one epoch (in MB):
> |  | infosedd | gan-dime | hd-dime | kl-dime | minde | mine | nwj | smile |
> | --- | --- | --- | --- | --- | --- | --- | --- | --- |
> | $MI=10, D=10$ | 346.37 | 645.81 | 645.81 | 645.81 | 440.51 | 645.81 | 645.81 | 645.81 |
> | $MI=20, D=20$ | 687.22 | 1301.98 | 1301.98 | 1301.98 | 876.44 | 1301.98 | 1301.98 | 1301.98 |
> | $MI=30, D=30$ | 1005.66 | 1913.15 | 1913.15 | 1913.15 | 1286.35 | 1913.15 | 1913.15 | 1913.15 |
> | $MI=40, D=40$ | 1345.94 | 2564.33 | 2564.33 | 2564.33 | 1726.25 | 2564.32 | 2564.32 | 2564.32 |
> | $MI=50, D=50$ | 1666.30 | 3180.50 | 3180.50 | 3180.50 | 2138.41 | 3180.49 | 3180.49 | 3180.49 |
>
> and runtime of one epoch (in seconds):
> |  | infosedd | gan-dime | hd-dime | kl-dime | minde | mine | nwj | smile |
> | --- | --- | --- | --- | --- | --- | --- | --- | --- |
> | $MI=10, D=10$ | 2.40 | 2.89 | 2.90 | 2.86 | 2.16 | 2.89 | 2.96 | 2.94 |
> | $MI=20, D=20$ | 3.34 | 3.99 | 4.02 | 3.98 | 2.62 | 4.03 | 4.03 | 4.00 |
> | $MI=30, D=30$ | 4.51 | 5.56 | 5.54 | 5.57 | 3.37 | 5.57 | 5.57 | 5.61 |
> | $MI=40, D=40$ | 5.96 | 7.34 | 7.33 | 7.36 | 4.00 | 7.39 | 7.42 | 7.37 |
> | $MI=50, D=50$ | 7.06 | 8.71 | 8.69 | 8.71 | 4.68 | 8.76 | 8.73 | 8.72 |
>
> As we can see, Infosedd requires around 20% less time to run and around 50% less memory compared to variational competitors, offering a better performance-accuracy trade-off for applications like sufficient statistics learning that require frequent MI estimates.

---

### Official Review · Reviewer_jnfn · 2025-11-05

**Soundness:** 3
**Presentation:** 3
**Contribution:** 3
**Rating:** 6
**Confidence:** 5

**Summary:**

This work extends the recent trend of information theoretic diffusion based estimators to discrete data.
The approach is "decomposable" in the sense that a single training run to estimate the mutual information of a vector allows for the estimation of that of any subvector.
Empirical evaluation is extensive, both accuracy of estimation and quality of learned representations are evaluated in multiple environment of increasing scale.

**Strengths:**

+ The paper is well written.
+ Advanced stochastic calculus concepts are explained simply, making the paper accessible to wider audience.
+ Estimation of mutual information on discrete data is promising and useful.
+ Amortized inference on subsequence is bound to be useful in many situations.
+ Evaluation on genomic data is a beautiful application of the estimator.

**Weaknesses:**

Most of the weaknesses have to do with the empirical evaluation.
+ Table 1: Missing error bars.
+ Evaluations are mostly presented as a graph which can be misleading.
+ Table 2: Correlation with human metrics only report INFO-SEDD and no baselines.
+ Overall evaluation feels somewhat indirect. While having the merit of being grounded in real life problems,  the empirical evaluation could be made stronger by evaluating the estimator itself as an objective for a clearly defined machine learning downstream tasks.

**Questions:**

Can do author suggest more direct application of Info-SEDD to machine learning tasks?

---

> ### Author Response · Authors · 2025-11-20
>
> We thank the reviewer for the careful assessment of our work.
> We really appreciate the positive comments about the quality and clarity of our work, as well as the remarks on the practical usefulness of our proposed method. Next, we address the reviewer's concerns point-by-point.
>
> 1. **Missing error bars**. We ran additional experiments using 10 different seeds and updated Table 1 as follows:
>
> |  | infosedd | gan-dime | hd-dime | kl-dime | minde | mine | nwj | smile |
> | --- | --- | --- | --- | --- | --- | --- | --- | --- |
> | $MI=10, D=10$ | 9.92 $\pm$ 0.12 | 12.15 $\pm$ 0.89 | 9.73 $\pm$ 0.43 | 8.38 $\pm$ 0.90 | 14.01 $\pm$ 2.91 | 10.21 $\pm$ 6.33 | 6.16 $\pm$ 2.11 | 12.83 $\pm$ 0.95 |
> | $MI=20, D=20$ | 20.02 $\pm$ 0.21 | 22.09 $\pm$ 1.75 | 12.65 $\pm$ 1.07 | 7.51 $\pm$ 0.56 | 26.98 $\pm$ 3.16 | 8.82 $\pm$ 0.80 | 6.50 $\pm$ 0.84 | 23.11 $\pm$ 1.41 |
> | $MI=30, D=30$ | 29.83 $\pm$ 0.54 | 20.74 $\pm$ 1.75 | 11.72 $\pm$ 2.69 | 7.02 $\pm$ 0.43 | 31.08 $\pm$ 4.33 | 7.41 $\pm$ 1.23 | 6.35 $\pm$ 0.34 | 21.79 $\pm$ 1.08 |
> | $MI=40, D=40$ | 39.11 $\pm$ 0.65 | 19.64 $\pm$ 1.33 | 11.68 $\pm$ 0.94 | 6.52 $\pm$ 0.32 | 33.97 $\pm$ 3.32 | 6.91 $\pm$ 0.66 | 6.24 $\pm$ 0.59 | 20.13 $\pm$ 1.27 |
> | $MI=50, D=50$ | 47.77 $\pm$ 1.18 | 17.27 $\pm$ 1.46 | 10.47 $\pm$ 1.12 | 6.41 $\pm$ 0.62 | 32.60 $\pm$ 3.93 | 7.21 $\pm$ 1.14 | 5.95 $\pm$ 0.31 | 18.97 $\pm$ 1.05 |
>
> Our updated results confirm that Infosedd outperforms all alternative methods.
> We observe that, in addition to improved accuracy, infosedd also has lower and more stable variance compared to other baselines, which is a highly desirable property. Indeed, variational estimators suffer from a variance that scales with the ground-truth MI value (see theorem 2 in [1]).
>
> [1] Song et al. Understanding the limitations of vaiational mutual information estimators, International Conference on Learning Representations, 2020
>
> 2. **Evaluations as graph vs tables**. We agree with the reviewer that tables offer a more detailed picture.
> This is why we have included, in our original submission, tabular results in Appendix C.
> Our plan is to improve those tables: we will make them more compact, and we will highlight the best estimates compared to the ground truth.
> Moreover, we will explicitly reference them in the captions of the graphs presented in the main paper so that readers interested in detailed measures can easily find such auxiliary tables.
>
> 3. **Missing baselines for human correlation metrics**. We agree with the reviewer: our additional results, including the best competitors shown below, are very helpful in highlighting the properties of our mutual information (MI) estimator.
>
> New table for Pearson correlation:
> |             |   infosedd-j |   infosedd-c |   smile |   kl-dime |   hd-dime |
> |:------------|-------------:|-------------:|--------:|-----------:|-----------:|
> | consistency |        0.550 |        0.740 |  -0.074 |      0.214 |      0.331 |
> | fluency     |        0.455 |        0.679 |  -0.162 |      0.194 |      0.281 |
> | relevance   |        0.288 |        0.411 |  -0.149 |      0.076 |     -0.145 |
> | coherence   |       -0.091 |        0.209 |  -0.367 |      0.170 |     -0.243 |
> | overall     |        0.322 |        0.568 |  -0.221 |      0.193 |      0.063 |
>
>
> New table for Kendall-Tau correlation:
> |             |   infosedd-j |   infosedd-c |   smile |   kl-dime |   hd-dime |
> |:------------|-------------:|-------------:|--------:|-----------:|-----------:|
> | consistency |        0.486 |        0.505 |  -0.105 |      0.429 |     -0.029 |
> | fluency     |        0.153 |        0.134 |   0.019 |      0.096 |      0.057 |
> | relevance   |        0.219 |        0.200 |  -0.067 |      0.048 |     -0.067 |
> | coherence   |        0.048 |        0.105 |  -0.238 |      0.105 |     -0.162 |
> | overall     |        0.238 |        0.219 |  -0.086 |      0.067 |     -0.048 |
>
> Our new results show that the alternative methods do not manifest any correlation with human metrics.

---

> ### Author Response · Authors · 2025-11-20
>
> 4. **Direct applications of infosedd?**. We agree with the reviewer: a direct use of MI for downstream tasks is an important topic to study and assess. In the literature, for continuous data, works such as [2, 3, 4] show that information-theoretic measures can be directly used as auxiliary loss functions for model finetuning, for example. Moreover, recent advances in Reinforcement Learning for discrete diffusion models [5] offer an exciting opportunity to use information metrics as a reward signal for downstream tasks.
> In our work, we gave a different spin to our experimental protocols: resting on the strong results achieved by Infosedd on our benchmarks, we decided to broaden the scope of our evaluation and reach the broader audience of practitioners who might be interested in using information estimation in their fields of research.
> Nevertheless, we agree that the integration of MI in downstream tasks is a natural next step for infosedd, and we will explore this direction in our future work.
>
> [2] Bounoua et al. Learning to Match Unpaired Data with Minimum Entropy Coupling, International Conference on Machine Learning 2025
>
> [3] Wang et al. Information Theoretic Text-to-Image Alignment, International Conference on Learning Representations 2024
>
> [4] Belghazi et al. Mutual information neural estimation, International Conference on Machine Learning 2018
>
> [5] Zekri et al. Fine-tuning discrete diffusion models with policy gradient methods, arXiv 2025

---

### Meta-Review · Area_Chair_evfD · 2026-01-07

**Summary:**

This paper proposes a discrete diffusion-based approach that establishes a connection between mutual information estimation and generative modeling for high-dimensional discrete data. Reviewers  jnfn, C1dg, and ps2M provide positive assessments. Concerns raised by reviewer gDGn have been substantively addressed in the rebuttal. Overall, review comments acknowledge the paper's novelty and its demonstrable improvement over prior state-of-the-art results. Based on these assessments, I agree with reviewers and recommend acceptance.

**Reviewer Concerns:**

Most reviewer concerns have heen addressed in the rebuttal, such as concerns regarding the differences from MINDE, analysis of estimation bias, and clarity of notations.

**Reviewer Scores:**

There is no clear evidence to suggest that any reviewer would have changed their score after a full discussion.

---

### Decision · Program_Chairs · 2026-01-26

Accept (Poster)